# Protein-based condensation mechanisms drive the assembly of RNA-rich P granules

**Helen Schmidt, Andrea Putnam, Dominique Rasoloson, Geraldine Seydoux\***

HHMI and Department of Molecular Biology and Genetics, Johns Hopkins University School of Medicine, Baltimore, United States

**Abstract** Germ granules are protein-RNA condensates that segregate with the embryonic germline. In *Caenorhabditis elegans* embryos, germ (P) granule assembly requires MEG-3, an intrinsically disordered protein that forms RNA-rich condensates on the surface of PGL condensates at the core of P granules. MEG-3 is related to the GCNA family and contains an N-terminal disordered region (IDR) and a predicted ordered C-terminus featuring an HMG-like motif (HMGL). We find that MEG-3 is a modular protein that uses its IDR to bind RNA and its C-terminus to drive condensation. The HMGL motif mediates binding to PGL-3 and is required for co-assembly of MEG-3 and PGL-3 condensates in vivo. Mutations in HMGL cause MEG-3 and PGL-3 to form separate condensates that no longer co-segregate to the germline or recruit RNA. Our findings highlight the importance of protein-based condensation mechanisms and condensate-condensate interactions in the assembly of RNA-rich germ granules.

## Introduction

In animals with germ plasm, specification of the germline depends on the segregation of maternal RNAs and proteins (germline determinants) to the primordial germ cells. Germline determinants assemble in germ granules, micron-sized dense assemblies that concentrate RNA and RNA-binding proteins (*Jamieson-Lucy and Mullins, 2019*; *Marnik and Updike, 2019*; *Seydoux, 2018*; *Trcek and Lehmann, 2019*). Superficially, germ granules resemble RNA-rich condensates that form in the cytoplasm of somatic cells, including P bodies and stress granules. In recent years, much progress has been made in our understanding of stress granule assembly with the realization that stress granules resemble liquid condensates that assemble by liquid-liquid phase separation (LLPS). LLPS is a thermodynamic process that causes interacting molecules to dynamically partition between a dense condensed phase and a more dilute phase (e.g., the cytoplasm) (*Banani et al., 2017*; *Mitrea and Kriwacki, 2016*). Low-affinity-binding interactions, often involving disordered and RNA-binding domains, are sufficient to drive LLPS of proteins and RNA in reconstituted systems (*Lin et al., 2015*; *Molliex et al., 2015*; *Zagrovic et al., 2018*). The ability of RNA to phase separate in the absence of proteins in vitro has also been proposed to contribute to RNA granule assembly in vivo, especially in the case of stress granules, which arise under conditions of general translational arrest (*Tauber et al., 2020*; *Van Treeck et al., 2018*). An emerging model is that the combined action of many low-affinity interactions between RNA molecules and multivalent RNA-binding proteins creates RNA-based protein networks that drive LLPS (*Guillén-Boixet et al., 2020*; *Sanders et al., 2020*; *Yang et al., 2020*; *Zhang et al., 2015*).

Unlike the dynamic condensates assembled by LLPS in vitro, germ granules are not well-mixed, single-phase liquid droplets. High-resolution microscopy has revealed that germ granules are heterogenous assemblies of dynamic and less dynamic condensates that co-assemble but do not fully mix. For example, *Drosophila* germ granules contain non-dynamic RNA clusters embedded in dynamic, protein-rich condensates (*Little et al., 2015*; *Niepielko et al., 2018*; *Trcek et al., 2015*). Germ granules in zebrafish and *Xenopus* are built on an amyloid-like scaffold that organizes mRNAs

**\*For correspondence:**
gseydoux@jhmi.edu

in nonoverlapping, transcript-specific zones (*Boke et al., 2016*; *Fuentes et al., 2018*; *Roovers et al., 2018*). The mechanisms that bring together condensates with different material properties and their contribution to RNA recruitment in germ granules are not well understood.

In this study, we examine the assembly of P granules, germ granules in *Caenorhabditis elegans*. At the core of P granules are liquid condensates assembled by PGL proteins. PGL-1 and PGL-3 are self-dimerizing, RGG domain proteins that readily form condensates able to recruit other P granule components, such as the VASA-related RNA helicase GLH-1 (*Aoki et al., 2016*; *Hanazawa et al., 2011*; *Saha et al., 2016*; *Updike et al., 2011*). PGL condensates exist in germ cells throughout oogenesis and are maternally inherited by the embryo. In newly fertilized zygotes, the surface of PGL condensates becomes covered by smaller condensates assembled by MEG-3 and MEG-4, two homologous intrinsically disordered proteins (*Wang et al., 2014*). Unlike PGL condensates, MEG-3 condensates resist dilution and salt challenge, consistent with a gel-like material (*Putnam et al., 2019*). (In this study, we use the term condensate to refer to concentrated protein assemblies that self-assemble without implying a mechanism for assembly, which could involve aggregation, LLPS, or other mechanisms, and may or may not include RNA.) During zygote polarization, MEG-3 and MEG-4 condensates enrich with other germ plasm components in the posterior cytoplasm (*Putnam et al., 2019*; *Smith et al., 2016*; *Wang et al., 2014*). This relocalization correlates with preferential growth of MEG-coated PGL droplets in the posterior and dissolution of 'naked' PGL droplets in the anterior side (*Brangwynne et al., 2009*; *Smith et al., 2016*). In addition to PGL and MEG co-assemblies, P granules also concentrate specific maternal transcripts (*Parker et al., 2020*; *Seydoux and Fire, 1994*). A survey of mRNAs that immunoprecipitate with PGL-1 and MEG-3 suggests that MEG-3 is most directly responsible for recruiting mRNAs to P granules (*Lee et al., 2020*). MEG-3 binds to ~500 maternal mRNAs, including transcripts coding for germline determinants. Recruitment of mRNAs to P granules ensures their preferential segregation to the primordial germ cells. Embryos lacking MEG-3 and MEG-4 do not localize PGL droplets, do not condense P granule-associated mRNAs, and display partially penetrant (30%) sterility (*Lee et al., 2020*; *Wang et al., 2014*).

To understand how MEG-3 coordinates PGL and RNA condensation, we used genome editing of the *meg-3* locus and reconstitution experiments in vitro to define functional domains in MEG-3. We find that MEG-3 is a bifunctional protein with separate domains for RNA recruitment and protein condensation. We identify a predicted ordered motif (HMGL) required for binding to PGL-3 in vitro that is essential to build MEG-3/PGL-3 co-assemblies that recruit RNA in vivo. The MEG-3 IDR binds RNA and enriches MEG-3 in germ plasm but is not sufficient on its own to assemble RNA-rich condensates. Our observations highlight the importance of condensation driven by protein-protein interactions in the assembly of germ granules.

## Results

### The MEG-3 C-terminus is the primary driver of MEG-3 condensation in zygotes

IUPred2A (*Mészáros et al., 2018*) predicts in the MEG-3 sequence a N-terminal domain with high disorder and a C-terminal domain with lower disorder separated by a boundary region with mixed order/disorder (aa544–698) (*Figure 1A, B*). The C-terminus contains a predicted ordered 44 amino acid sequence (aa700–744) with homology to the HMG-like-fold found in the GCNA family of intrinsically disordered proteins (*Figure 1C*). Like MEG-3, GCNA family members contain long N-terminal disordered domains, but these do not share sequence homology with the MEG-3 IDR (*Carmell et al., 2016*). To test the functionality of MEG-3 domains in vivo, we used CRISPR genome editing to create four MEG-3 derivatives at the endogenous locus: MEG-3$_{Cterm}$ (aa545–862); MEG-3$_{IDR}$ (aa1–544); MEG-3$_{698}$, an extended version of MEG-3$_{IDR}$ terminating right before the HMG-like motif; and MEG-3$_{HMGL-}$, a full-length MEG-3 variant with alanine substitutions in four conserved residues in the HMG-like motif (*Figure 1B, C*). (We also constructed a MEG-3$_{Cterm}$ (aa545–862) variant with mutations in the HMG-like motif, but this variant was not expressed at sufficiently high levels for analysis.) The MEG-3 variants were created in a *C. elegans* line where the *meg-4* locus was deleted to avoid possible complementation by MEG-4, a close MEG-3 paralog. To allow visualization of MEG-3 protein by immunofluorescence, each variant (and wild-type *meg-3*) was tagged with a

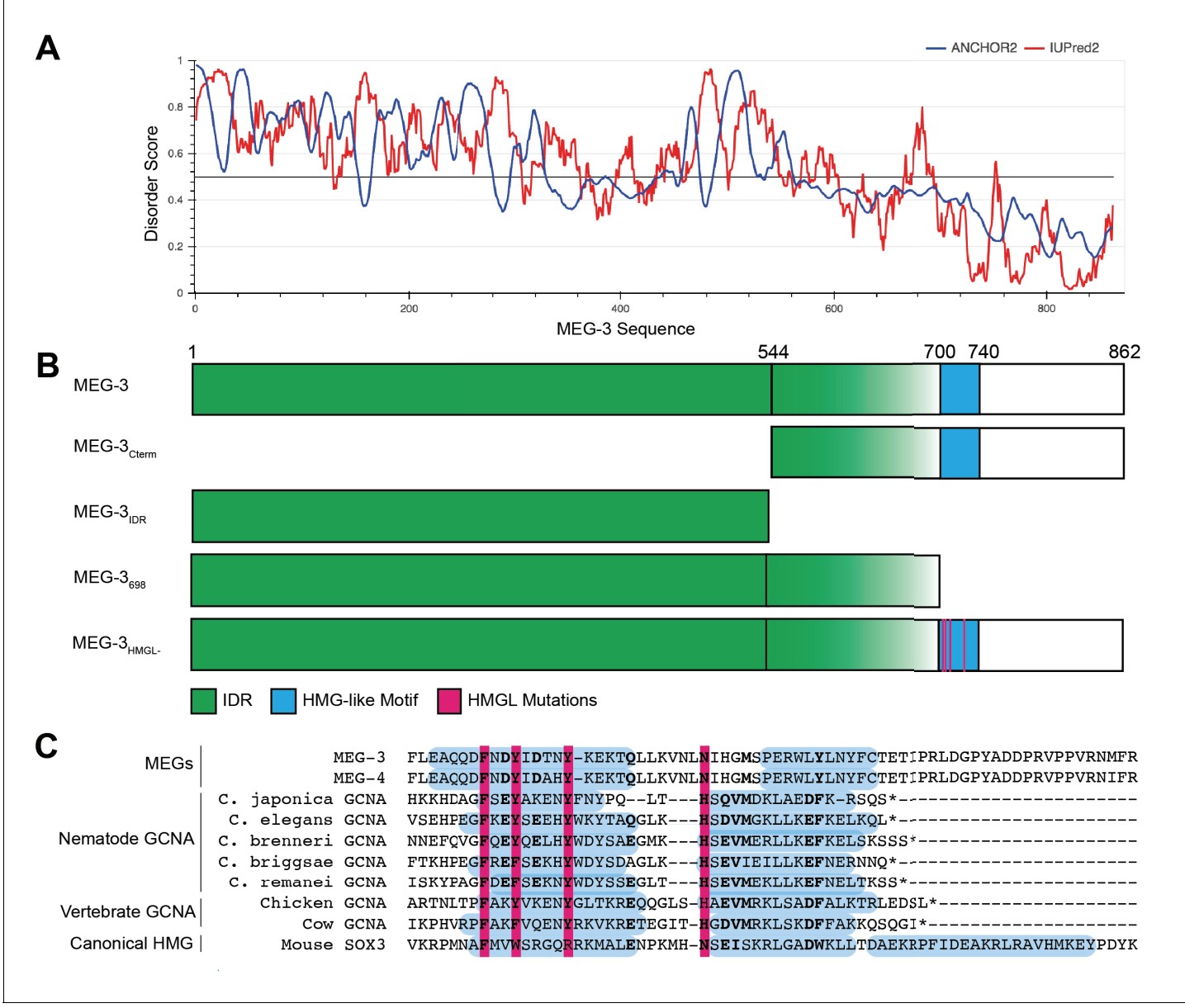

**Figure 1.** Domain organization of MEG-3. (**A**) MEG-3 amino acid sequence (N- to C-terminus) on the X-axis is plotted against disorder score on the Y-axis as predicted by ANCHOR2 (blue) and IUPred2 (red) (*Mészáros et al., 2018*) with a range from 0 to 1, where 1 is the most disordered. (**B**) Schematics of wild-type MEG-3 and four MEG-3 variants analyzed in this study. Amino acid positions are aligned with (**A**). The disordered region (green) and HMG-like motif (blue) are indicated. Magenta bars (alanine substitutions) correspond to four conserved residues in the HMG-like motif shaded in magenta in (**C**). (**C**) Alignment of the HMG-like motif in MEG-3 and MEG-4 with the HMG-like motif in GCNA proteins (*Carmell et al., 2016*) and the canonical HMG box of mouse SOX3. Amino acids predicted to form alpha-helices are highlighted in blue (*Drozdetskiy et al., 2015*). Bold indicates positions with >70% amino acid similarity. Magenta bars indicate residues mutated to alanine in MEG-3$_{HMGL-}$.

C-terminal OLLAS peptide (*Figure 2A*). We avoided the use of fluorescent tags as fluorescent tags have been reported to affect the behavior of proteins in P granules (*Uebel and Phillips, 2019*).

As reported previously for untagged MEG-3 (*Wang et al., 2014*), MEG-3 tagged with OLLAS could be detected diffusively in the cytoplasm and in condensates (*Figure 2A*). Before polarization, MEG-3 was uniformly distributed throughout the zygote. After polarization, MEG-3 in the cytoplasm and in condensates became enriched in the posterior half of the zygote destined for the germline blastomere $P_1$ ('germ plasm'). MEG-3 continued to segregate preferentially with P blastomeres in subsequent divisions ($P_1$ through $P_4$) (*Figure 2A*).

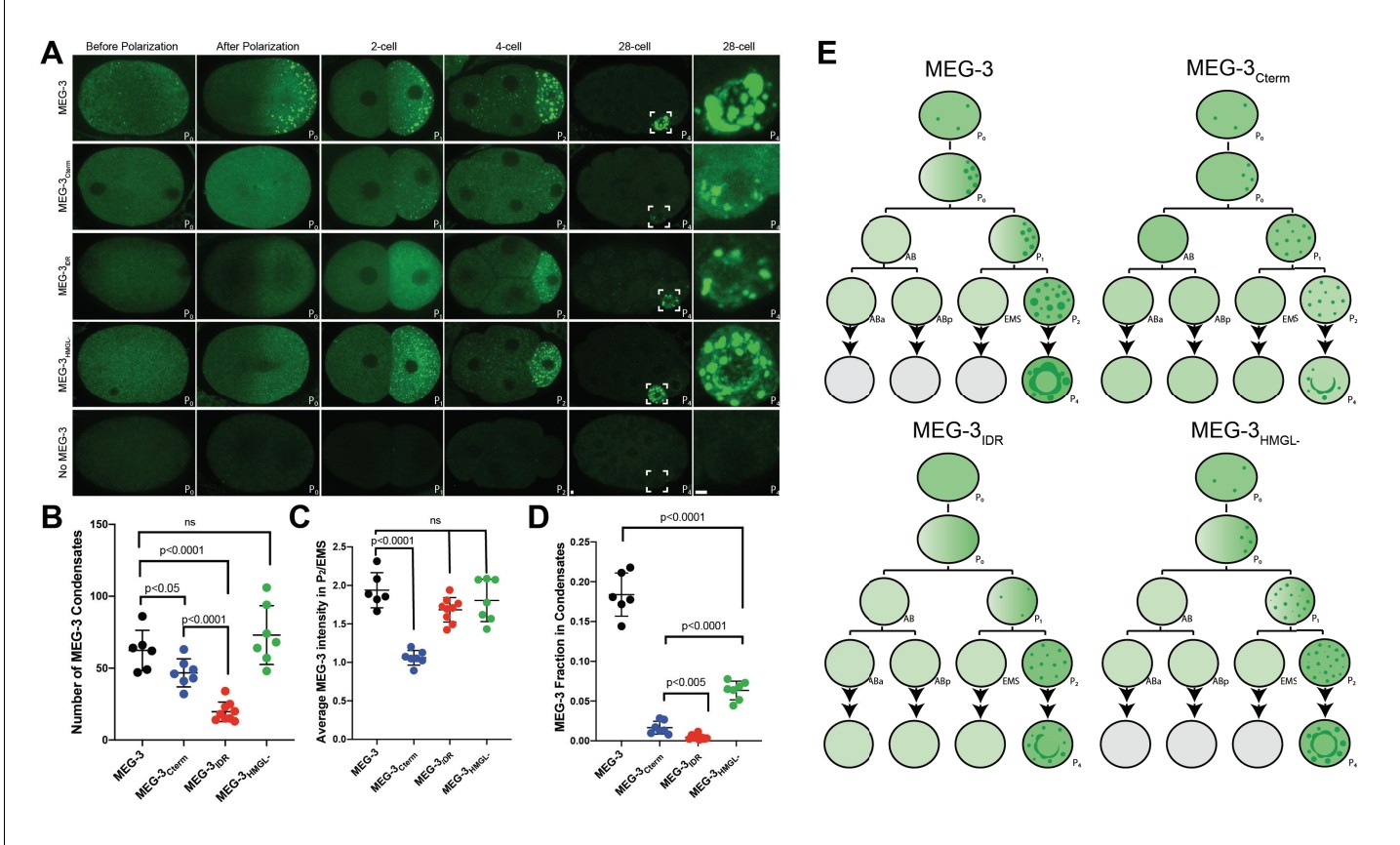

**Figure 2.** Localization of wild-type MEG-3 and variants in early embryos. (**A**) Representative photomicrographs of embryos immunostained for OLLAS and expressing the indicated OLLAS-tagged MEG-3 derivatives. Last row shows *meg-3 meg-4* embryos as negative control for OLLAS staining. Images are representative of stages indicated above each column. Before and after polarization are one-cell stage zygotes, other stages are indicated by the total number of cells in each stage. The name of the P (germ) blastomere is indicated in the bottom right of each image. A minimum of three embryos from two independent experiments were analyzed for each stage. Scale bars are 1 μm. All images are maximum projections normalized to same fluorescent intensity range except for the last column showing high-magnification views of $P_4$ from the 28-cell stage image adjusted to highlight MEG-3 granules. (**B**) Scatterplot showing the number of MEG-3 condensates in the $P_2$ blastomere in embryos expressing the indicated MEG-3 derivatives. Each dot represents an embryo. (**C**) Scatterplot showing enrichment of MEG-3 in the $P_2$ blastomere over the somatic blastomere (EMS), calculated by dividing the average intensity in $P_2$ by the average intensity in EMS. Each dot represents an embryo also included in the analysis shown in (**B**). (**D**) Scatterplot showing the fraction of the MEG-3 signal localized to condensates over total signal in $P_2$. Each dot represents an embryo also included in the analysis in (**B**). (**E**) Summary of MEG-3 (green) distribution derived from data presented in (**A**). Each row corresponds to a different stage as in (**A**), starting with unpolarized zygote, polarized zygote, 2-cell, 4-cell, and 28-cell stage. Horizontal lines denote one-cell division, arrows indicate multiple divisions. Note that wild-type MEG-3 and MEG-3_HMGL- are rapidly turned over in somatic cells after the four-cell stage (gray cells) as shown in *Figure 2—figure supplement 1B, C*.

The online version of this article includes the following source data and figure supplement(s) for figure 2:

**Source data 1.** Condensation and enrichment of MEG-3 in four-cell embryos.

**Figure supplement 1.** Additional characterization of wild-type MEG-3 and variants in embryos.

All four MEG-3 variants exhibited unique localization patterns distinct from wild-type. MEG-3_IDR enriched in posterior cytoplasm and segregated preferentially to P blastomeres but did not appear robustly in condensates until the four-cell stage (P_2 blastomere, *Figure 2A*). MEG-3_698 behaved identically to MEG-3_IDR (*Figure 2—figure supplement 1A*). MEG-3_Cterm did not enrich asymmetrically in the cytoplasm but formed condensates in the zygote posterior and continued to form condensates only in P blastomeres despite being present in the cytoplasm of all cells (*Figure 2A*). MEG-3_HMGL- behaved most similarly to wild-type MEG-3 enriching in the zygote posterior and forming condensates as early as the one-cell stage, although the condensates appeared smaller at all stages (*Figure 2A*).

For each MEG-3 derivative, we quantified the number of condensates and the degree of enrichment in the P blastomere ($P_2$) over somatic blastomeres and in condensates over the cytoplasm at the four-cell stage. Wild-type MEG-3 and MEG-3$_{HMGL-}$ formed a similar number of condensates, while MEG-3$_{Cterm}$ formed fewer and MEG-3$_{IDR}$ the least in the four-cell stage (*Figure 2B*). The MEG-3$_{Cterm}$ did not enrich in the $P_2$ blastomere, whereas MEG-3$_{IDR}$ and MEG-3$_{HMGL-}$ enriched as efficiently as wild-type (*Figure 2C*). Finally, none of MEG-3 derivatives enriched in condensates as efficiently as wild-type (*Figure 2D*).

After the four-cell stage, the low levels of wild-type MEG-3 and MEG-3$_{HMGL-}$ inherited by somatic blastomeres were rapidly cleared. In contrast, MEG-3$_{IDR}$ and MEG-3$_{Cterm}$ persisted in somatic blastomeres at least until the 28-cell stage (*Figure 2—figure supplement 1B*). Western analyses revealed that MEG-3 and MEG-3$_{HMGL-}$ accumulate to similar levels, whereas MEG-3$_{IDR}$ and MEG-3$_{Cterm}$ were more abundant in mixed-stage embryo lysates, consistent with slower turnover in somatic lineages (*Figure 2—figure supplement 1C*).

The condensation, segregation, and turnover patterns of MEG-3, MEG-3$_{IDR}$, MEG-3$_{Cterm}$, and MEG-3$_{HMGL-}$ are summarized in *Figure 2E*. From this analysis, we conclude that (1) the MEG-3 IDR is necessary and sufficient for enrichment of cytoplasmic MEG-3 in germ plasm, (2) the MEG-3 C-terminus is necessary and sufficient to assemble MEG-3 condensates in germ plasm starting in the zygote stage, (3) the HMG-like motif enhances, but is not essential for, condensation, and (4) both the C-terminus and the IDR are required for timely turnover of MEG-3 in somatic lineages.

## Co-assembly of MEG-3/PGL-3 condensates in vivo is driven by the MEG-3 C-terminus and requires the HMGL motif

MEG-3 and MEG-4 are required redundantly to localize PGL condensates to the posterior of the zygote for preferential segregation to the P lineage (*Smith et al., 2016*; *Wang et al., 2014*). To examine the distribution of PGL condensates relative to MEG-3 condensates, we utilized the KT3 and OLLAS antibodies for immunostaining of untagged endogenous PGL-3 and OLLAS-tagged MEG-3. In embryos expressing wild-type MEG-3, MEG-3 and PGL-3 co-localize in posterior condensates that are segregated to the $P_1$ blastomere (*Figure 3A*). In embryos lacking *meg-3* and *meg-4*, PGL-3 condensates distributed throughout the cytoplasm of the zygote and segregated equally to AB and $P_1$ (*Figure 3A*). We observed a similar pattern in embryos expressing MEG-3$_{IDR}$, MEG-3$_{698}$, and MEG-3$_{HMGL-}$ indicating that none of these MEG-3 derivatives are sufficient to localize PGL condensates (*Figure 3A*, *Figure 3—figure supplement 1A*). In contrast, in embryos expressing MEG-3$_{Cterm}$, PGL-3 condensates preferentially assembled in $P_1$, although they were smaller and fewer than in wild-type (*Figure 3A, B*). Embryos expressing MEG-3$_{Cterm}$ enriches PGL-3 in $P_1$, though not as efficiently as wild-type, while PGL-3 is not enriched in *meg-3 meg-4*, or embryos expressing MEG-3$_{IDR}$ or MEG-3$_{HMGL-}$ (*Figure 3C*). In wild-type 28-cell stage embryos, PGL-3 condensates are highly enriched in $P_4$. No such enrichment was observed in embryos expressing the MEG-3$_{Cterm}$ or any other MEG-3 variant (*Figure 3—figure supplement 1*). We conclude that the MEG-3$_{Cterm}$ is sufficient to enrich PGL-3 condensates in P blastomeres in early stages, but not sufficient to support robust PGL-3 localization through $P_4$.

Wild-type MEG-3 condensates associate closely with the surface of PGL condensates (*Putnam et al., 2019*; *Wang et al., 2014*). With the resolution afforded by immunostaining, this configuration appears as co-localized MEG and PGL puncta in fixed embryos (*Wang et al., 2014*, *Figure 3B*). We found that PGL-3 condensates co-localized with MEG-3$_{Cterm}$ condensates (37/37 PGL-3 condensates scored in $P_1$; *Figure 3B*) as in wild-type. In contrast, we observed no such co-localization with MEG-3$_{IDR}$ or MEG-3$_{HMGL-}$. The MEG-3$_{IDR}$ is mostly cytoplasmic and forms only rare condensates in $P_1$. We occasionally observed PGL condensates with an adjacent MEG-3$_{IDR}$ condensate (5/19 PGL-3 condensates scored in $P_1$, *Figure 3B*), but these were not co-localized. Unlike the MEG-3$_{IDR}$, MEG-3$_{HMGL-}$ forms many condensates in $P_1$, although these tended to be smaller than wild-type (*Figure 3A, B*). Still, although we occasionally observed PGL condensates with an adjacent MEG-3$_{HMGL-}$ condensate (12/30 PGL-3 condensates scored in $P_1$; *Figure 3B*), we never observed fully overlapping PGL/MEG-3$_{HMGL-}$ co-condensates. We conclude that, despite forming many condensates in P blastomeres, MEG-3$_{HMGL-}$ condensates do not associate efficiently with, and do not support the localization of, PGL-3 condensates.

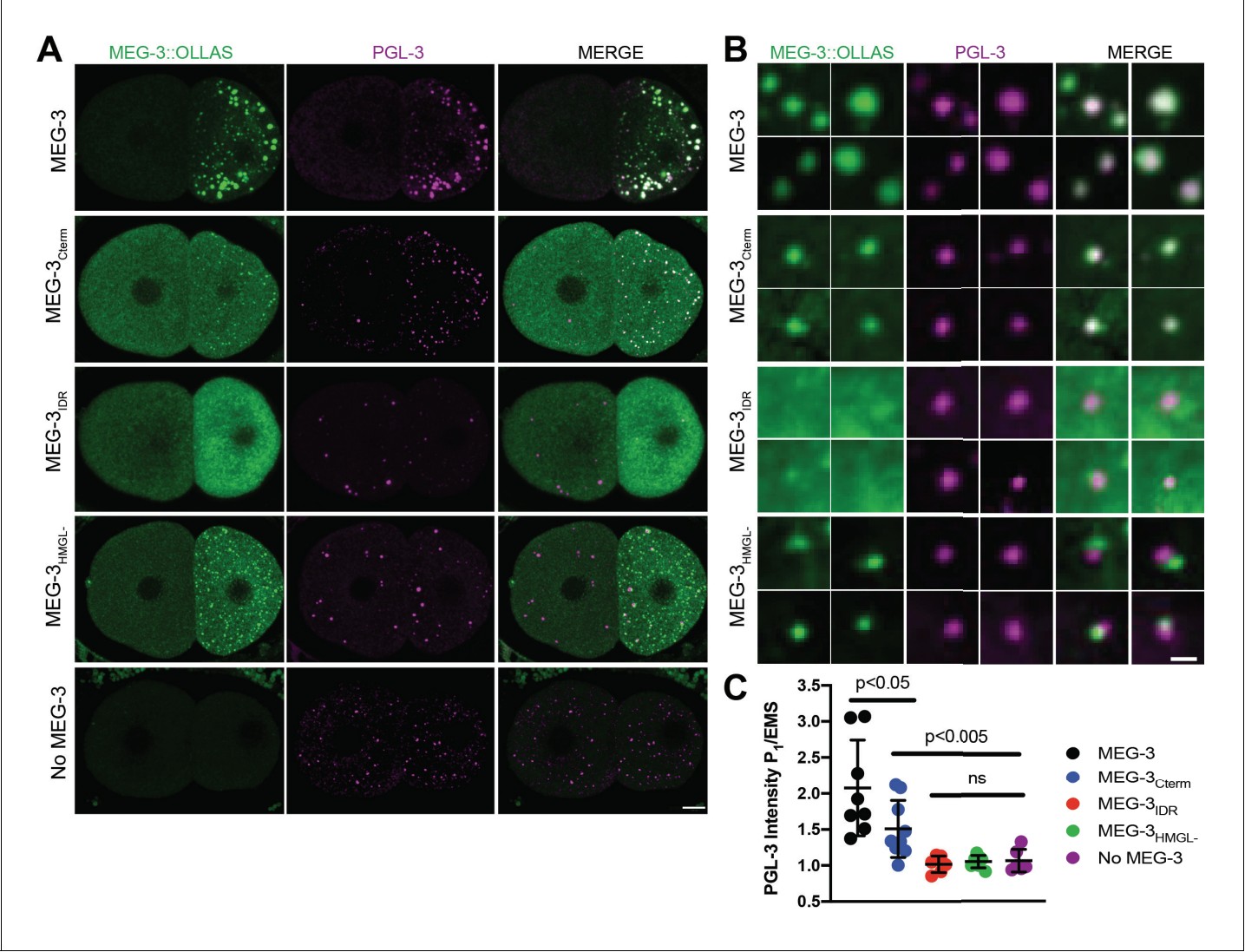

**Figure 3.** Localization of PGL-3 relative to wild-type MEG-3 and variants in two-cell embryos. (**A**) Representative photomicrographs of two-cell embryos expressing the indicated MEG-3 mutants and immunostained for MEG-3 (anti-OLLAS antibody) and PGL-3 (anti-PGL-3 antibody). Scale bar is 5 μm. (**B**) High-magnification photomicrographs of individual MEG-3/PGL-3 assemblies in embryos expressing the indicated MEG-3 derivatives. White color in the merge indicates overlap. Scale bar is 1 μm. (**C**) Scatterplot of the enrichment of PGL-3 in $P_1$ calculated by dividing the average intensity in $P_1$ by the average intensity in the somatic blastomere (AB). Each dot represents an embryo.

The online version of this article includes the following source data and figure supplement(s) for figure 3:

**Source data 1.** Enrichment of PGL-3 in $P_1$ in embryos expressing wild-type MEG-3 and variants.

**Figure supplement 1.** Localization of PGL-3 relative to wild-type MEG-3 and variants in $P_4$ blastomeres.

### Efficient recruitment of *Y51F10.2* mRNA to P granules requires the MEG-3 IDR, C-terminus, and HMG-like motif

MEG-3 recruits mRNAs to P granules by direct binding that traps mRNA into the non-dynamic MEG-3 condensates (*Lee et al., 2020*). To determine which MEG-3 domain is required for mRNA recruitment to MEG-3 condensates in vivo, we performed in situ hybridization against the MEG-3-bound mRNA *Y51F10.2*. Prior to polarization, *Y51F10.2* is uniformly distributed throughout the zygote cytoplasm (*Figure 4A*). *Y51F10.2* becomes progressively enriched in P granules starting in the late one-cell stage and forms easily detectable micron-sized foci by the four-cell stage (*Lee et al., 2020*; *Figure 4A*). In contrast, in *meg-3 meg-4* embryos, *Y51F10.2* remains uniformly distributed in the cytoplasm at all stages. Strikingly, we observed the same failure to assemble *Y51F10.2* foci in

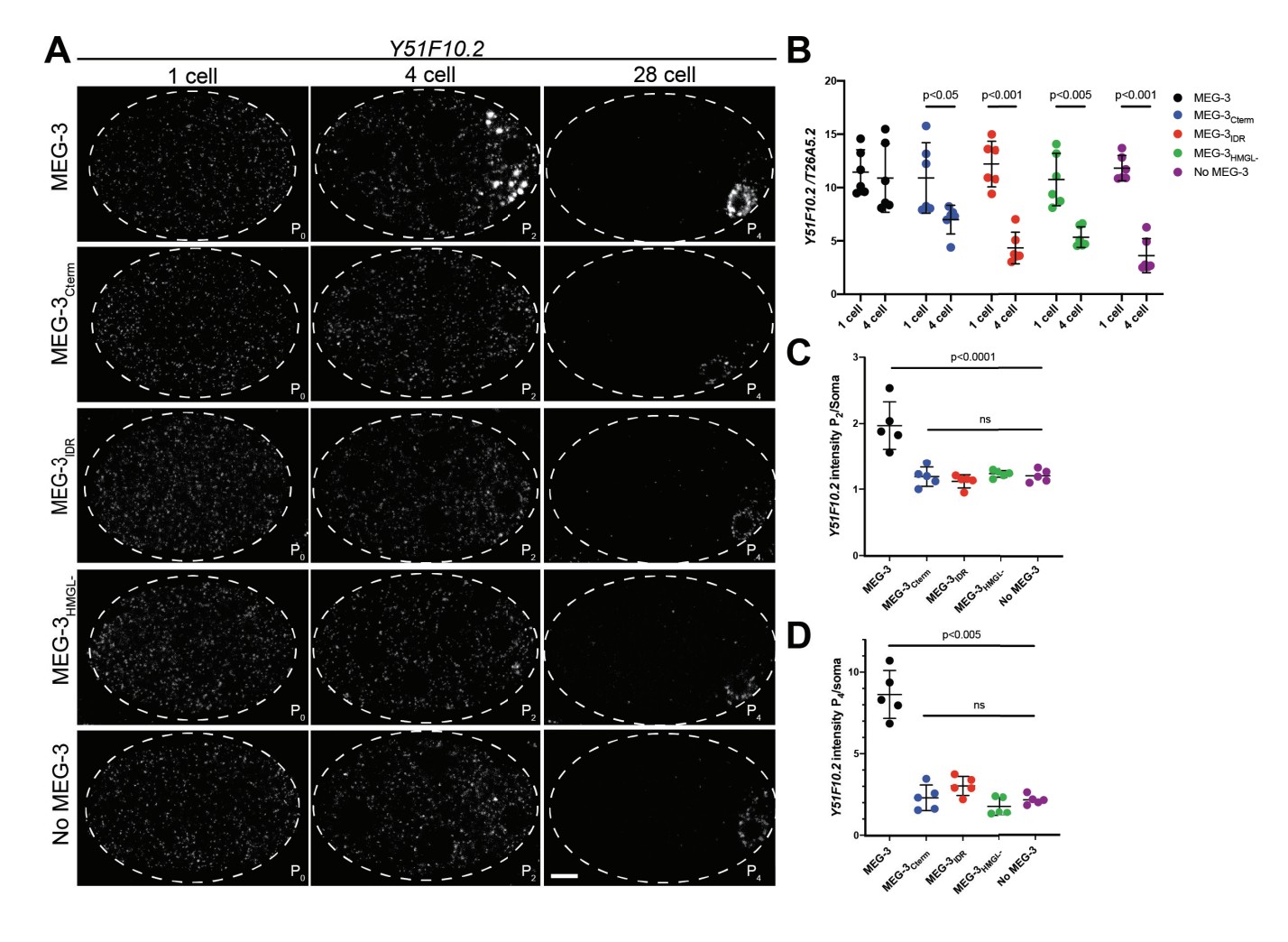

**Figure 4.** Distribution of *Y51F10.2* mRNA in embryos expressing wild-type MEG-3 and variants. (A) Representative photomicrographs of single confocal slices of fixed embryos expressing the indicated MEG-3 variants and hybridized to fluorescent probes complementary to the P granule-enriched mRNA *Y51F10.2* (white signal). (B) Scatterplot showing the ratio of *Y51F10.2* to *T26A5.2* mRNA signal in $P_0$ and $P_2$ embryos expressing the indicated MEG-3 derivatives. Each dot represents an embryo. See *Figure 4—figure supplement 1* for *T26A5.2* mRNA localization and levels. (C) Scatterplot showing enrichment of *Y51F10.2* mRNA in $P_2$ relative to somatic blastomeres in embryos expressing the indicated MEG-3 derivatives. Each dot represents an embryo (Materials and methods). (D) Scatterplot showing enrichment of *Y51F10.2* mRNA in $P_4$ relative to somatic blastomeres in embryos expressing the indicated MEG-3 derivatives. Each dot represents an embryo (Materials and methods).

The online version of this article includes the following source data and figure supplement(s) for figure 4:

**Source data 1.** *Y51F10.2* mRNA levels in embryos expressing wild-type MEG-3 and variants.

**Figure supplement 1.** Distribution of *T26A5.2* in embryos expressing wild-type MEG-3 and variants.

**Figure supplement 2.** Distribution of *nos-2* mRNA in embryos expressing wild-type MEG-3 and variants.

**Figure supplement 3.** Distribution of polyadenylated mRNA in embryos expressing wild-type MEG-3 and variants.

embryos expressing MEG-3$_{IDR}$, MEG-3$_{Cterm}$, and MEG-3$_{HMGL-}$ (*Figure 4A*). This was surprising since all three MEG-3 variants form visible condensates by the four-cell stage (*Figure 3A*).

To characterize the fate of *Y51F10.2* transcripts in *meg-3 meg-4* mutants, we compared the intensity of the *Y51F10.2* in situ hybridization signal relative to a control RNA (*T26A5.2*) in one-cell and four-cell stage embryos (*Figure 4B*, Materials and methods, *Figure 4—figure supplement 1*). In wild-type, *Y51F10.2* RNA levels do not change significantly from the one-cell to the four-cell stage. In contrast, in *meg-3 meg-4* embryos, *Y51F10.2* levels decreased by ~50% by the four-cell stage, despite starting at levels similar to wild-type in the one-cell stage. This finding is consistent with

RNAseq results, which indicated lower levels of P granule mRNAs in *meg-3 meg-4* embryos (*Lee et al., 2020*). We observed a similar loss of *Y51F10.2* RNA in embryos expressing MEG-3$_{IDR}$, MEG-3$_{Cterm}$, and MEG-3$_{HMGL-}$ (*Figure 4B*). We repeated this analysis with a second MEG-3-bound mRNA, *nos-2*. Similar to *Y51F10.2*, *nos-2* levels remained constant from the one-cell to four-cell stage in embryos expressing wild-type MEG-3, and decreased in *meg-3 meg-4* embryos and embryos expressing MEG-3$_{IDR}$, MEG-3$_{Cterm}$, and MEG-3$_{HMGL-}$ (*Figure 4—figure supplement 2*). These results suggest that failure to recruit *Y51F10.2* and *nos-2* mRNAs into granules leads to their premature degradation.

After the four-cell stage, as has been reported for other maternal RNAs (*Baugh et al., 2003*; *Seydoux and Fire, 1994*), *Y51F10.2* is rapidly turned over in somatic blastomeres. At the four-cell stage, *Y51F10.2* mRNA levels were approximately twofold higher P$_2$ than in somatic blastomeres in wild-type embryos, and ~1.2-fold higher in *meg-3 meg-4* embryos and in embryos expressing MEG-3$_{IDR}$, MEG-3$_{Cterm}$, and MEG-3$_{HMGL-}$ (*Figure 4C*). By the 28-cell stage, in wild-type embryos, *Y51F10.2* levels were ~10-fold higher in the germline founder cell P$_4$ compared to somatic blastomeres. In contrast, in *meg-3 meg-4* embryos, *Y51F10.2* mRNA levels were only approximately twofold enriched over somatic levels. Similarly, in embryos expressing the MEG-3$_{IDR}$, MEG-3$_{Cterm}$, and MEG-3$_{HMGL-}$, *Y51F10.2* enrichment in P$_4$ averaged around approximately twofold (*Figure 4A, D*).

Enrichment of mRNAs in P granules can also be detected using an oligo-dT probe to detect poly-adenylated mRNAs (*Seydoux and Fire, 1994*). In wild-type 28-cell stage embryos, a concentrated poly-A signal is detected around the nucleus of the P$_4$ blastomere. Poly-A signal intensity in P$_4$ is ~1.8-fold higher than that observed in somatic blastomeres (*Figure 4—figure supplement 3A, B*). This enrichment was not detected in *meg-3 meg-4* mutants. In *meg-3 meg-4* mutants, and in embryos expressing the three MEG-3 derivatives, poly-A signal intensity was similar between the somatic and P$_4$ blastomeres. The lack of poly-A enrichment in P$_4$ was particularly striking in the case of MEG-3$_{IDR}$ and MEG-3$_{HMGL-}$ since those variants assemble robust perinuclear condensates in P$_4$ (*Figure 2A*).

Failure to efficiently segregate and stabilize maternal mRNAs in P blastomeres has been linked to the partial penetrance maternal-effect sterility (~30%) of *meg-3 meg-4* mutants (*Lee et al., 2020*). We observed similar levels of sterility in hermaphrodites derived from mothers expressing the MEG-3$_{IDR}$, MEG-3$_{Cterm}$, and MEG-3$_{HMGL-}$ (*Figure 4—figure supplement 3C*). We conclude that the MEG-3 C-terminus, IDR, and HMG-like motif are all required for efficient mRNA recruitment to P granules, which in turn is required for enrichment and stabilization in the P lineage and robust germ cell fate specification.

## The MEG-3 IDR is necessary and sufficient for RNA binding in vitro

We showed previously that MEG-3 readily condenses with RNA in vitro (*Lee et al., 2020*). To analyze the properties of MEG-3 variants in vitro, we expressed and purified His-tagged full-length MEG-3, MEG-3$_{Cterm}$, MEG-3$_{IDR}$, MEG-3$_{698}$, MEG-3$_{HMGL-}$, and MEG-3$_{Cterm\ HMGL-}$, a MEG-3$_{Cterm}$ variant with a mutated HMGL motif (same alanine substitutions as in MEG-3$_{HMGL-}$). We first tested each variant for its ability to bind RNA. Using fluorescence polarization and gel shift assays, we previously showed that the MEG-3 IDR binds an RNA oligo (poly-U(30)) with near nanomolar affinity in vitro (*Smith et al., 2016*). We repeated these observations using a filter binding assay where proteins are immobilized on a filter to minimize possible interference due to condensation of MEG-3 in solution (Materials and methods, *Figure 5A*). Consistent with previous observations (*Smith et al., 2016*), we found that MEG-3$_{IDR}$ and MEG-3$_{698}$ exhibit high affinity for poly-U(30) (*Figure 5A*, *Figure 5—figure supplement 1B*). Wild-type MEG-3 also bound poly-U(30) efficiently albeit at a lower affinity compared to MEG-3$_{IDR}$ and MEG-3$_{698}$. In contrast, MEG-3$_{Cterm}$ exhibited negligible RNA binding (*Figure 5A*). HMG domains are common in DNA-binding proteins and have been shown to mediate protein:nucleic acid interactions in vivo (*Genzor and Bortvin, 2015*; *Reeves, 2001*; *Thapar, 2015*). MEG-3$_{HMGL-}$ bound to RNA as efficiently as full-length MEG-3, indicating that the HMGL motif does not contribute to RNA binding in MEG-3 (*Figure 5A*).

In vivo, MEG-3 binds to maternal mRNAs including *nos-2* (*Lee et al., 2020*). To examine the affinity of MEG-3 for *nos-2* RNA, we repeated the filter binding assay with labeled poly-U(30) using unlabeled *nos-2* RNA as a competitor (*Figure 5B*). The competition assay revealed that full-length MEG-

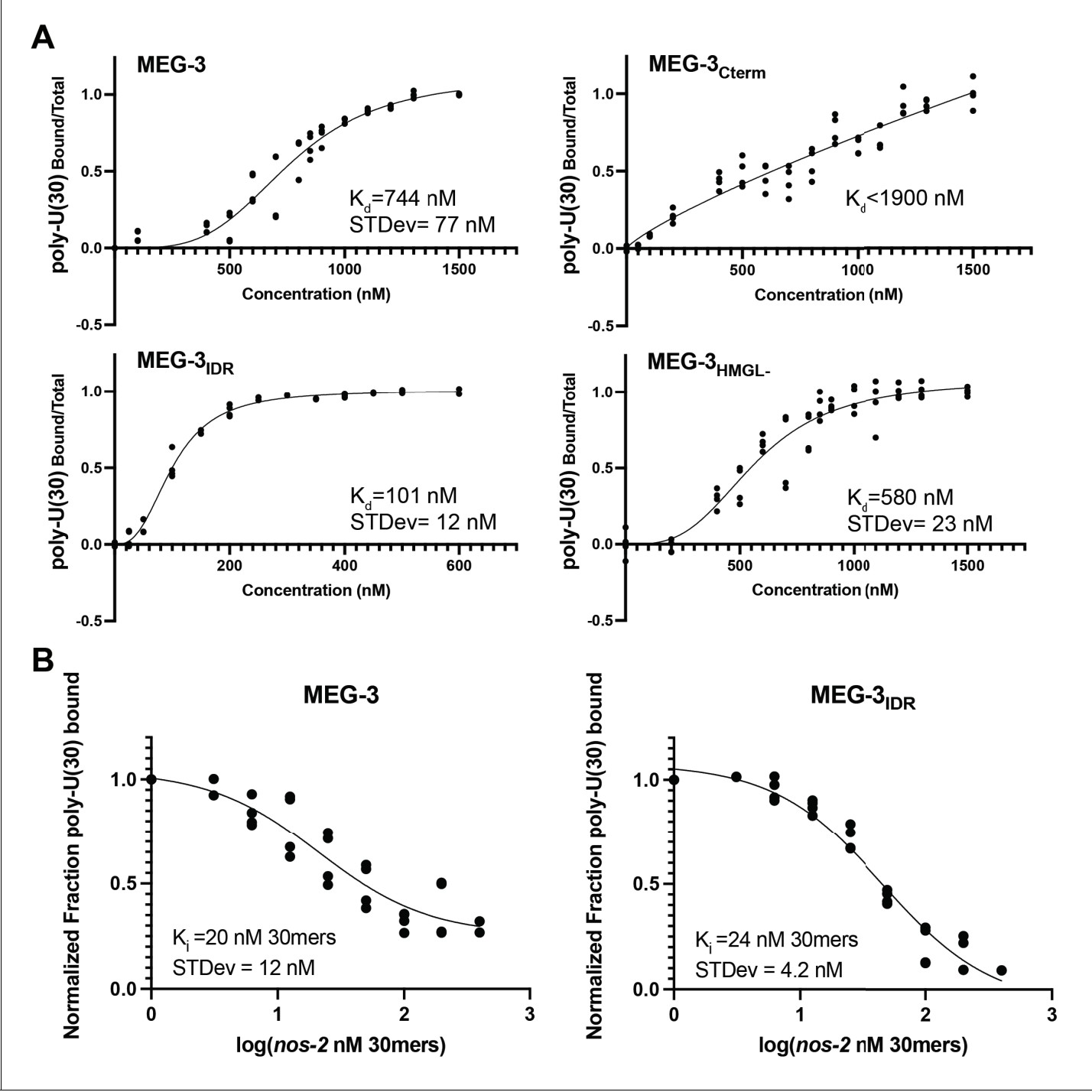

**Figure 5.** In vitro RNA binding of wild-type MEG-3 and variants. See *Figure 5—figure supplement 1* for an SDS-PAGE gel of the purified proteins used in *Figures 5*, *6*, *7*. (**A**) 30U RNA-binding curves for MEG-3, MEG-3$_{Cterm}$, MEG-3$_{IDR}$, and MEG-3$_{HMGL-}$. Protein concentration is plotted on the X-axis. The ratio of bound poly-U(30) RNA to total, normalized to the ratio at the maximum concentration, is plotted on the Y-axis. Each dot represents a replicate at a given concentration. The average K$_d$ and standard deviation were calculated from four replicate curves fit independently to a specific binding with Hill slope model (Materials and methods). (**B**) Competitive *nos-2* RNA-binding curves for MEG-3 and MEG-3$_{IDR}$. The log of *nos-2* RNA concentration is plotted on the X-axis. The ratio of bound poly-U(30) to total RNA, normalized to the ratio in the absence of *nos-2*, is plotted on the Y-axis. Each dot represents a replicate at a given concentration. The average K$_i$ and standard deviation were calculated from four replicate curves fit independently to a one-site competitive binding model (Materials and methods).

The online version of this article includes the following source data and figure supplement(s) for figure 5:

*Figure 5 continued on next page*

*Figure 5 continued*

**Source data 1.** RNA binding of MEG-3 and variants in vitro.
**Figure supplement 1.** In vitro purified MEG-3 and variants.

3 and MEG-3$_{IDR}$ bind to *nos-2* RNA with the same high affinity. We conclude that the MEG-3 IDR is necessary and sufficient for RNA binding.

## The MEG-3 C-terminus is the primary driver of MEG-3 condensation in vitro

To determine which regions of MEG-3 are required for condensation in vitro, MEG-3 and variants were trace-labeled with covalently attached fluorophores and examined for condensate formation by microscopy. MEG-3 condensation is sensitive to salt and RNA concentration with RNA having a strong solubilizing influence especially in low salt (*Lee et al., 2020*). We tested four conditions varying RNA and salt and keeping MEG-3 concentration constant at 150 nM near the physiological range (*Figure 6A*, *Figure 6—figure supplement 1*). At that concentration, MEG-3 and variants were all soluble in low salt/high RNA condensation buffer (50 mM NaCl to 80 ng/µL *nos-2* RNA) (*Figure 6—figure supplement 1*). Under higher salt conditions (150 mM NaCl salt and 20 ng/µL or 80 ng/µL *nos-2* RNA), MEG-3$_{IDR}$ and MEG-3$_{698}$ remained mostly soluble forming only rare condensates (*Figure 6A*, *Figure 6—figure supplements 1*, *2*). In contrast, full-length MEG-3 formed robust condensates under those conditions. MEG-3$_{C-term}$ also formed condensates, albeit with lower efficiency compared to MEG-3 (*Figure 6A*, *Figure 6—figure supplement 1*). The MEG-3$_{Cterm}$ condensates were also approximately twofold less efficient at recruiting RNA compared to full-length MEG-3 and MEG-3$_{IDR}$ (*Figure 6A*, *Figure 6—figure supplement 1B*). We conclude that, as observed in vivo, the C-terminus is the primary driver of MEG-3 condensation. The MEG-3 C-terminus is not sufficient, however, for maximum condensation or RNA recruitment, which additionally require the IDR.

Surprisingly, under all conditions, MEG-3$_{HMGL-}$ and MEG-3$_{CtermHMGL-}$ behaved identically to MEG-3 and MEG-3$_{Cterm}$, respectively (*Figure 6*, *Figure 6—figure supplement 1*), indicating that the HMGL motif is dispensable for RNA recruitment and condensation in vitro. HMGL was required for RNA recruitment and efficient condensation in vivo (*Figure 4*), suggesting that our in vitro conditions do not fully reproduce in vivo conditions (see below).

## Formation of a MEG-3 condensate layer on PGL-3 condensates requires the MEG-3 C-terminus and does not require the IDR or RNA in vitro

When combined in condensation buffer, MEG-3 and PGL-3 form co-condensates that resemble the architecture of P granules in vivo, with the smaller MEG-3 condensates (~100 nm) forming a dense layer on the surface of the larger PGL-3 condensates (*Putnam et al., 2019*; *Figure 6B*, *Figure 6—figure supplement 3A*). We found that the MEG-3$_{Cterm}$ formed co-condensates with PGL-3 that were indistinguishable from those formed by wild-type MEG-3 (*Figure 6—figure supplement 3A, B*). The MEG-3$_{IDR}$, in contrast, mixed homogenously with the PGL-3 phase as previously reported (*Putnam et al., 2019*, *Figure 6—figure supplement 3A*). MEG-3$_{HMGL-}$ and MEG-3$_{CtermHMGL-}$ behaved as MEG-3 and MEG-3$_{Cterm}$, respectively. In vivo, MEG-3$_{HMGL-}$ does not associate efficiently with PGL-3 condensates (*Figure 3B*), again suggesting that in vitro conditions do not reproduce the more stringent in vivo environment.

We repeated the co-condensation assays in the absence of RNA using a higher concentration of PGL to force PGL condensation in the absence of RNA. Under these conditions, MEG-3 and PGL-3 formed co-condensates that were indistinguishable from co-condensates assembled in the presence of RNA (compare right and left panels, *Figure 6—figure supplement 3A and B*). Again, the C-terminus was necessary and sufficient for co-assembly. MEG-3$_{IDR}$ homogenously mixed with the PGL-3 phase and did not form independent condensates. We conclude that condensation of MEG-3 on the surface of PGL condensates requires the MEG-3 C-terminus and does not require RNA in vitro.

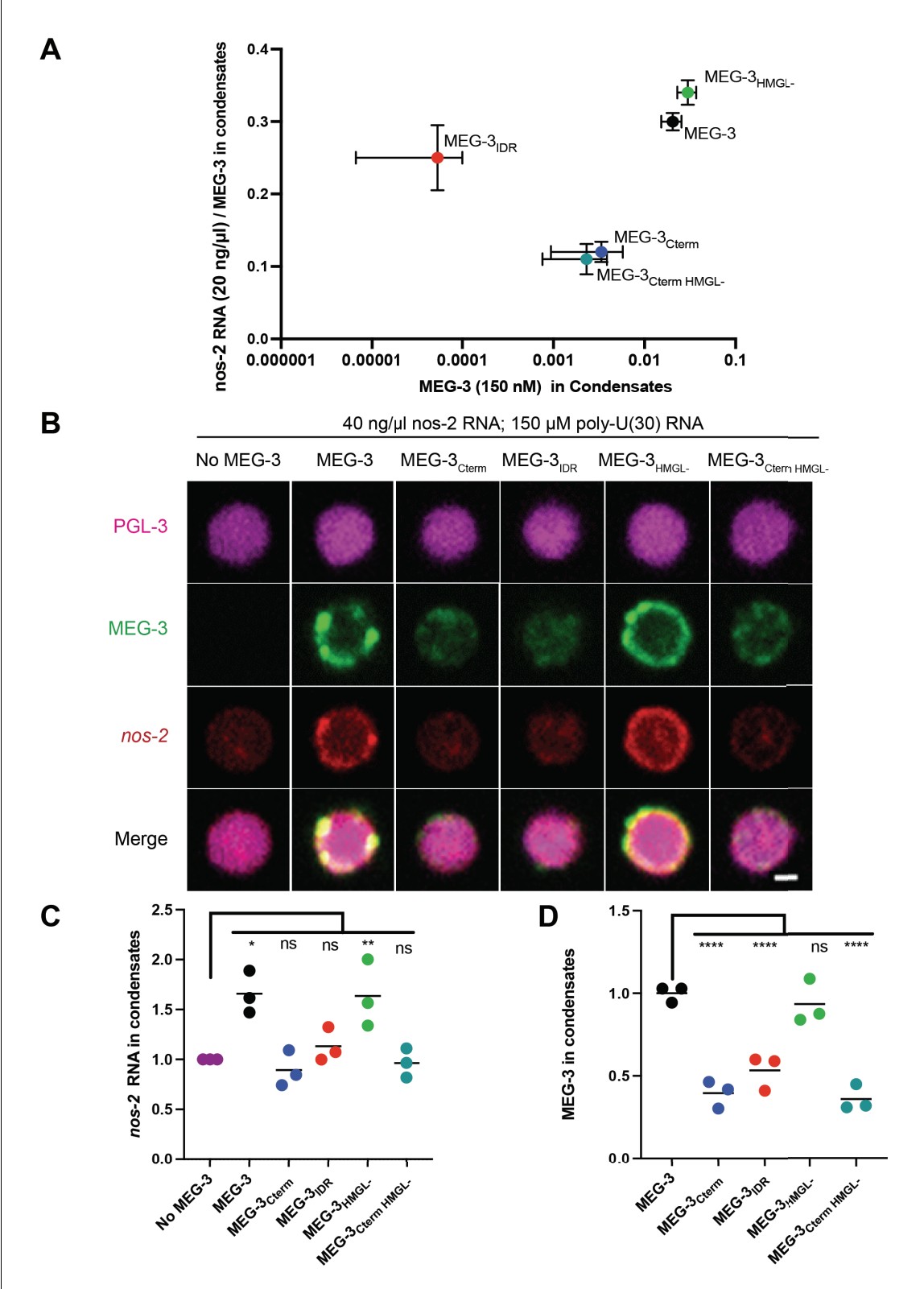

**Figure 6.** In vitro condensation of wild-type MEG-3 and variants. (**A**) Plot of the condensation of MEG-3 and MEG-3 derivatives incubated with 20 ng/µL *nos-2* mRNA and 150 nM salt. MEG-3 condensates were identified in ImageJ (Materials and methods). The total fluorescence intensity in condensates normalized to the total image intensity is plotted on the X-axis. The total intensity of RNA in condensates divided by the intensity of MEG-3 in condensates is plotted on the Y-axis. Each dot represents the mean of three replicates, and bars represent the standard deviation; three images from

*Figure 6 continued on next page*

*Figure 6 continued*

the same slide were quantified in each replicate. Representative photomicrographs and values at additional NaCl and RNA conditions are shown in *Figure 6—figure supplement 1*. (B) Representative photomicrographs of 150 nM MEG-3 and MEG-3 derivatives (trace-labeled with Alexa647) incubated for 30 min with 2.5 µM PGL-3 (trace-labeled with Dylight488), 40 ng/µL *nos-2* RNA (trace-labeled with Alexa555), and 150 µM poly-U(30) RNA. Full-field photomicrographs and photomicrographs of MEG-3/PGL-3 co-condensates with either 20 ng/µL *nos-2* RNA or no RNA in *Figure 6—figure supplement 3*. (C) Scatterplot of *nos-2* RNA enrichment in PGL-3 condensates with and without MEG-3 or MEG-3 variants in the presence of 150 µM poly-U30 RNA. The intensity of *nos-2* fluorescence in condensates divided by the total image intensity and normalized to the No MEG-3 condition is plotted on the Y-axis. Each dot represents a replicate; three images from the same slide were quantified in each replicate. (D) Scatterplot of MEG-3 and MEG-3 variants enrichment in PGL-3 condensates in the presence of 40 ng/µL *nos-2* RNA and 150 µM 30U RNA. The intensity of MEG-3 fluorescence in condensates divided by the total image intensity and normalized to the wild-type MEG-3 condition is plotted on the Y-axis. Each dot represents a replicate; three images from the same slide were quantified in each replicate.

The online version of this article includes the following source data and figure supplement(s) for figure 6:

**Source data 1.** MEG-3 and variants condensation with and without PGL-3 in vitro.
**Figure supplement 1.** In vitro condensation of MEG-3 and variants at different RNA and salt concentrations.
**Figure supplement 2.** MEG-3$_{698}$ condensation in vitro.
**Figure supplement 3.** Co-condensation of MEG-3 and PGL-3 in vitro.

## Recruitment of *nos-2* RNA to MEG-3/PGL-3 co-condensates requires the MEG-3 IDR and C-terminus

To examine the ability of MEG-3 to recruit *nos-2* RNAs to MEG-3/PGL-3 co-condensates, we assembled the co-condensates in the presence of labeled 40 ng/µL labeled *nos-2* RNA and unlabeled 150 µM poly-U(30). Unlabeled poly-U(30) was necessary to prevent *nos-2* RNA from accumulating non-specifically in the PGL-3 phase. Under these high RNA conditions, all variants were recruited to PGL condensates, but only full-length MEG-3 (and MEG-3$_{HMGL-}$) formed a distinctive layer around PGL condensates that recruited RNA above the background level recruited to the PGL-3 phase (*Figure 6B–D*). MEG-3$_{IDR}$ and MEG-3$_{Cterm}$, in contrast, mixed with the PGL phase and did not enrich *nos-2* RNA above background levels (*Figure 6B, C*). These observations suggest that enrichment of *nos-2* RNA in MEG-3/PGL-3 co-condensates requires robust MEG-3 condensation driven by the combined action of the MEG-3 IDR and C-terminus.

## The HMG domain is required for MEG-3 binding to PGL proteins

We reported previously that MEG-3 binds directly to PGL-1, as determined in a GST-pull-down assay using partially purified recombinant proteins (*Wang et al., 2014*). HMG domains have been implicated in protein-protein interactions (*Reeves, 2001*; *Stros et al., 2007*; *Wilson and Koopman, 2002*). To examine whether the HMG domain is required to mediate MEG-3/PGL binding, we repeated the GST-pull-down assay using fusion proteins of GST::MEG-3$_{Cterm}$ and MBP::PGL-1 and PGL-3 (*Figure 7A*). GST::MEG-3$_{IDR}$ fusions were not expressed and thus could not be tested in this assay. We found that the GST::MEG-3$_{Cterm}$ binds efficiently to PGL-1 and PGL-3, but not to MBP or to an unrelated control protein PAA-1 (*Figure 7A, B*). A GST::MEG-3$_{Cterm}$ fusion with mutations in the HMG-like domain bound less efficiently to both PGL-1 and PGL-3 (*Figure 7A, B*). To complement the GST assay, we examined the ability of purified labeled PGL-3 to bind to beads coated with purified labeled MEG-3 derivatives (bead halo assay) (*Figure 7C*, *Figure 7—figure supplement 1A-C*). We found that PGL-3 is recruited efficiently to beads coated with MEG-3$_{C-term}$, but not to beads coated with MEG-3$_{IDR}$ or MEG-3$_{Cterm\ HMGL-}$ (*Figure 7C*, *Figure 7—figure supplement 1A*). We conclude that the MEG-3$_{C-term}$ binds directly to PGL-3 and that this interaction requires the HMGL motif. This finding provides a potential explanation for why the HMGL is required for co-assembly of MEG-3 and PGL-3 condensates in vivo but not in vitro. Direct binding between MEG-3 and PGL-3 molecules may be necessary to assemble co-condensates in the crowded in vivo environment and may not be required in vitro where no other proteins or condensates compete for binding to MEG-3 or PGL-3.

## Discussion

In this study, we have examined the function of MEG-3 in P granule assembly using genome editing in vivo and recombinant proteins in vitro. We found that the MEG-3 IDR binds RNA and the MEG-3

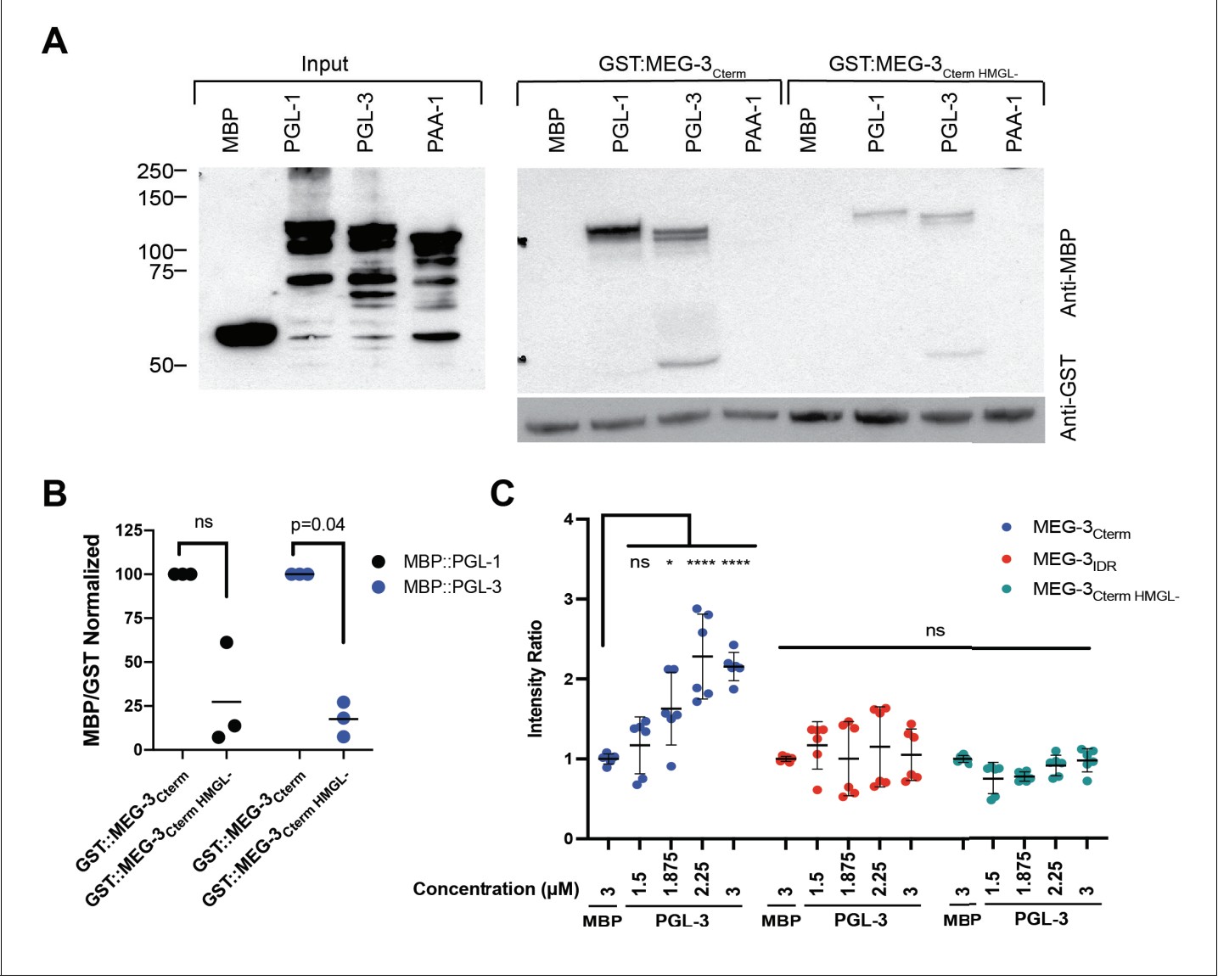

**Figure 7.** The HMGL motif is required for MEG-3 binding to PGL-3. (A) Analysis of GST::MEG-3$_{Cterm}$ and MBP::PGL-1 and MBP::PGL-3 interactions by GST-pull-down assay with MBP and MBP::PAA-1 as negative controls. Western blots of *Escherichia coli* lysates expressing the indicated MBP-fusions before (input) and after immobilization on magnetic beads with the indicated GST-fusions. Western blot of GST fusions is shown below. (B) Scatterplot of the ratio of the indicated MBP fusions to GST:MEG-3$_{Cterm\ HMGL-}$ normalized to the ratio of the same MBP fusion to GST:MEG-3$_{Cterm}$ from the same experiment. Each dot represents an independent pull-down experiment. p-values indicated above were calculated by a paired ratio t-test of the GST: MEG-3$_{Cterm}$ and GST:MEG-3$_{Cterm\ HMGL-}$ ratios before normalization. (C) Scatterplot of the ratio of PGL-3 (trace-labeled with Alexa555) to His-tagged MEG-3 derivatives (trace-labeled with Alexa647) immobilized on Nickel-NTA beads, normalized to the average ratio of MBP to MEG-3 derivatives (Materials and methods). Each dot represents an image containing multiple beads; two independent replicates with three images each were performed. SDS-PAGE gels of the protein inputs and representative photomicrographs of the beads in *Figure 7—figure supplement 1*. The online version of this article includes the following source data and figure supplement(s) for figure 7:

**Source data 1.** MEG-3 and variants binding to PGL-3.
**Figure supplement 1.** Bead halo assay for MEG-3/PGL-3 binding.

C-terminus drives condensation. In vitro, the IDR and C-terminus are sufficient to assemble MEG-3/ PGL-3 co-condensates that enrich RNA. In vivo, co-assembly additionally requires the HMGL motif, which mediates direct binding between MEG-3 and PGL-3. These findings (summarized in *Figure 8*) combined with prior analyses (*Putnam et al., 2019*; *Lee et al., 2020*) suggest the following model for P granule assembly: binding between PGL-3 and MEG-3 recruits MEG-3 to the surface of PGL-3

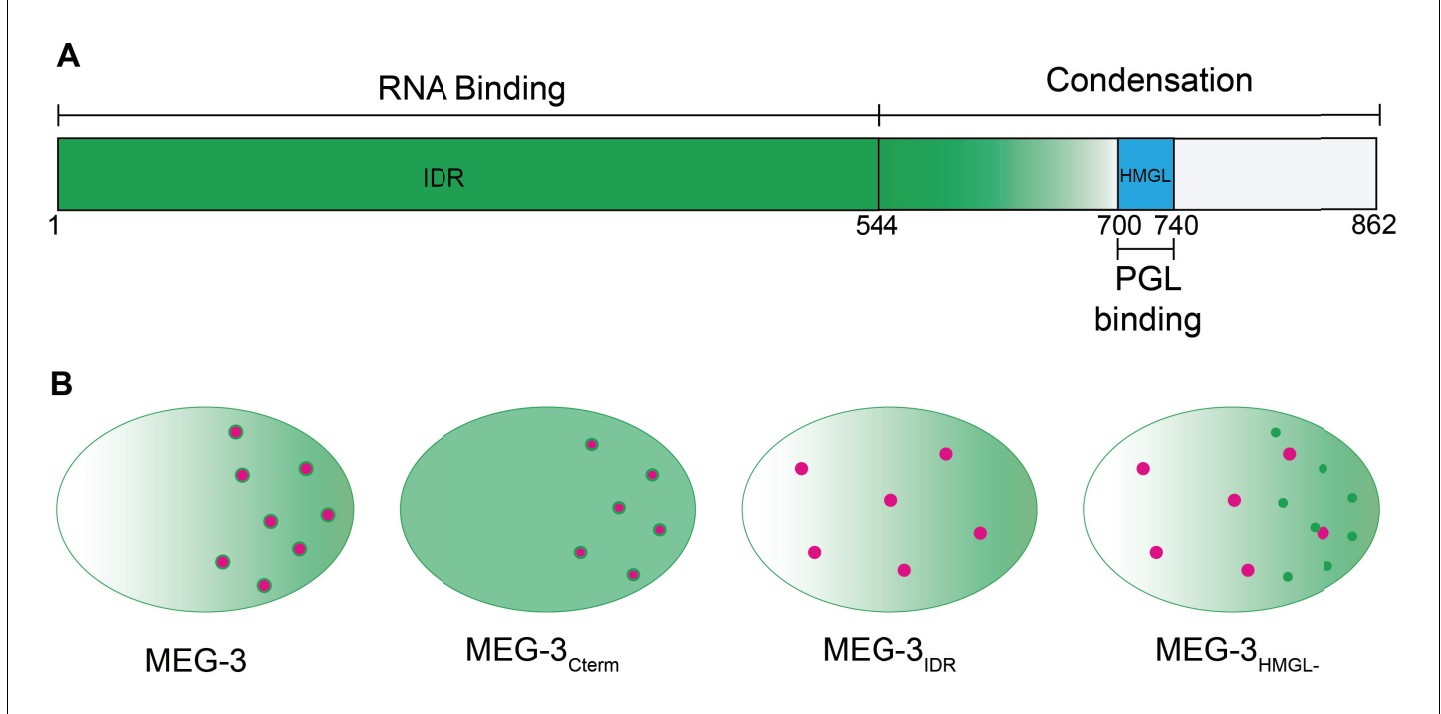

**Figure 8.** Model. (A) Schematic of MEG-3 function by region. The disordered region (green), ordered C-terminus (white), and HMG-like motif (blue) are indicated. (B) Schematics of one-cell zygotes showing distribution of MEG-3 (green) and PGL-3 (magenta). Wild-type MEG-3 forms robust condensates that recruit RNA and interact with, and enrich, PGL-3 condensates in posterior cytoplasm. MEG-3$_{Cterm}$ forms condensates that do not recruit RNA but still interact with, and enrich, PGL-3 condensates in posterior. MEG-3$_{IDR}$ localizes in posterior-rich cytoplasm but does not form condensates, and does not localize PGL-3. MEG-3$_{HMGL-}$ assembles condensates in posterior cytoplasm that do not recruit RNA and do not interact efficiently with, nor localize, PGL-3 condensates.

condensates in germ plasm and stimulates condensation of MEG-3 and MEG-3-bound RNAs. The MEG-3 layer protects mRNAs from degradation and stabilizes PGL-3 condensates in germ plasm ensuring their preferential segregation to the germline founder cell P$_4$.

## Assembly of MEG-3/PGL-3 co-condensates depends on the MEG-3 C-terminus and HMGL domain and does not require RNA

The MEG-3 C-terminus contains an HMG-like motif required to bind to PGL-3 and additional predicted low-disorder sequences that drive condensation by an unknown mechanism. The HMGL domain is not required for condensation in vitro but is required for maximal condensation efficiency in vivo. We suggest the HMGL domain enhances condensation indirectly in vivo by concentrating MEG-3 on the surface of PGL droplets, which stimulates condensation (*Putnam et al., 2019*). Docking of P bodies on stress granules has been proposed to involve RNA:RNA duplexes (*Tauber et al., 2020*). In contrast, we find that docking of MEG-3 condensates on PGL condensates does not require RNA in vitro and can occur in the absence of any visible RNA enrichment in vivo. Most strikingly, mutations in the HMGL domain that prevent binding between MEG-3 and PGL-3 molecules in solution prevent docking of MEG-3 and PGL condensates in vivo. Together, these observations suggest that MEG-3 condensation and co-assembly with PGL-3 condensates is driven primarily by protein-protein interactions and does not require RNA. We note that the HMGL domain is dispensable for co-assembly of MEG-3 and PGL-3 condensates in vitro, indicating that our condensation assay conditions do not fully reproduce the stringent environment of the cytoplasm.

## Efficient MEG-3/PGL co-assembly correlates with stabilization of PGL droplets in germ plasm

We previously reported that enrichment of PGL droplets to the posterior of the zygote requires *meg-3* (*Smith et al., 2016*; *Wang et al., 2014*). Our new findings suggest that this activity is linked to MEG-3's ability to associate stably with the PGL interface. MEG-3$_{Cterm}$, which is sufficient for MEG-3/PGL co-assembly, is sufficient to localize PGL in zygotes. Conversely, MEG-3$_{HMGL-}$ condensates, which do not interact stably with PGL condensates, fail to enrich PGL condensates in the posterior of zygotes. PGL localization involves preferential growth and dissolution of PGL droplets in the anterior and posterior, respectively. One possibility is that tight binding of MEG condensates lowers the surface tension of PGL droplets allowing MEG/PGL co-assemblies in the posterior to grow at the expense of the less stable, 'naked' PGL droplets in the anterior.

What enriches MEG-3 condensates in the posterior? We previously hypothesized that MEG-3 asymmetry is driven by a competition for RNA between the MEG-3 IDR and MEX-5, an RNA-binding protein that acts as an RNA sink in the anterior (*Smith et al., 2016*). Consistent with this hypothesis, the MEG-3 IDR is sufficient to enrich MEG-3 in posterior cytoplasm. Unexpectedly, however, we found that the MEG-3$_{Cterm}$ condenses preferentially in the zygote posterior despite uniform distribution in the cytoplasm, suggesting that additional mechanisms acting on the MEG-3 C-terminus contribute to MEG-3 regulation in space. Consistent with this view, a recent study examining MEG-3 dynamics by single-molecule imaging (*Wu et al., 2019*) found that the slowly diffusing MEG-3 molecules that populate the MEG-3 gradient in the cytoplasm represent a distinct population of MEG-3 molecules from those that associate with PGL droplets. We propose that MEG-3 asymmetry is sustained by two independent mechanisms: one acting on the MEG-3 C-terminus that biases condensation of MEG-3 in posterior cytoplasm and one acting on the MEG-3 IDR that enriches MEG-3 molecules in posterior cytoplasm. We speculate that the latter may serve to segregate high levels of MEG-3 to P blastomeres needed to support PGL asymmetry through the $P_4$ stage. Consistent with this view, the MEG-3$_{Cterm,}$ which does not enrich in a gradient, is not sufficient to localize PGL condensates in P blastomeres past the four-cell stage.

## MEG-3 condensation on PGL droplets creates a platform for RNA recruitment

The MEG-3 IDR binds RNA with high-affinity in vitro but is not sufficient to enrich RNA in vivo despite forming some condensates. RNA recruitment in vivo additionally requires the MEG-3 C-terminus including the HMG-like motif. These observations suggest that maximal MEG-3 condensation is essential to build a protein scaffold that can support stable RNA recruitment in vivo. Separate domains for RNA-binding and protein condensation have also been observed for other germ granule scaffolds. For example, the Balbiani body protein Xvelo uses a prion-like domain to aggregate and a separate RNA-binding domain to recruit RNA (*Boke et al., 2016*). Similarly, condensation of *Drosophila* Oskar does not require the predicted Oskar RNA-binding domain, although this domain augments condensation (*Kistler et al., 2018*). These observations parallel our findings with MEG-3 and contrast with recent findings reported for the stress granule scaffold G3BP. Condensation of G3BP in vitro requires RNA and two C-terminal RNA-binding domains. A N-terminal dimerization domain is also required but, unlike the prion-like domain of Xvelo or the C-terminus of MEG-3, is not sufficient to drive condensation on its own. Dimerization of G3BP is thought to enhance LLPS indirectly by augmenting the RNA-binding valency of G3BP complexes. G3BP also contains an inhibitory domain that gates its RNA-binding activity and condensation at low RNA concentrations. This modular organization ensures that G3BP functions as a sensitive switch that initiates LLPS when sufficient RNA molecules are available to cross-link G3BP dimers into a large network (*Guillén-Boixet et al., 2020*; *Yang et al., 2020*). Stress granules are transient structures that form under conditions of general translational arrest where thousands of transcripts are released from ribosomes. In contrast, germ granules are long-lived structures that assemble in translationally active cytoplasm and recruit only a few hundred specific transcripts (~500 in *C. elegans* embryos) (*Jamieson-Lucy and Mullins, 2019*; *Lee et al., 2020*; *Trcek and Lehmann, 2019*; *Updike and Strome, 2010*). One possibility is that protein-based condensation mechanisms may be better suited to assemble long-lived granules able to capture and retain rare transcripts. By concentrating IDRs with affinity for RNA, protein scaffolds could act as seeds for localized LLPS to amplify protein and RNA condensation. Consistent

with this view, IDRs have been observed to undergo spontaneous LLPS in cells when artificially tethered to protein modules that self-assemble into large multimeric structures (*Nakamura et al., 2019*). A challenge for the future will be to understand the mechanisms that regulate the assembly and disassembly of protein scaffolds at the core of germ granules.

# Materials and methods

**Key resources table**

| Reagent type (species) or resource | Designation | Source or reference | Identifiers | Additional information |
|---|---|---|---|---|
| Strain, strain background (*Caenorhabditis elegans*) | JH3477 | *Smith et al., 2016* | MEG-3::OLLAS *meg-4* deletion | *meg-3(ax3051)* *meg-4(ax3052)* |
| Strain, strain background (*C. elegans*) | JH3479 | *Smith et al., 2016* | MEG-3$_{IDR}$::OLLAS *meg-4* deletion | *meg-3(ax3056)* *meg-4(ax3052)* |
| Strain, strain background (*C. elegans*) | JH3517 | This study | MEG-3$_{698}$::OLLAS MEG-4::3xFLAG | *meg-3(ax4500)* *meg-4(ax2080)* |
| Strain, strain background (*C. elegans*) | JH3630 | This study | MEG-3$_{698}$::OLLAS *meg-4* deletion | *meg-3(ax4500)* *meg-4(ax3052)* |
| Strain, strain background (*C. elegans*) | JH3632 | This study | MEG-3(HMGL deletion)::OLLAS *meg-4* deletion | *meg-3(ax4501)* *meg-4(ax3052)* |
| Strain, strain background (*C. elegans*) | JH3861 | This study | MEG-3$_{HMGL-}$::OLLAS *meg-4* deletion | *meg-3(ax4502)* *meg-4(ax3052)* |
| Strain, strain background (*C. elegans*) | JH3420 | This study | MEG-3$_{Cterm}$::OLLAS MEG-4::3xFLAG | *meg-3(ax4503)* *meg-4(ax2080)* |
| Strain, strain background (*C. elegans*) | JH3553 | This study | MEG-3$_{Cterm}$::OLLAS *meg-4* deletion | *meg-3(ax4503)* *meg-4(ax4504)* |
| Strain, strain background (*C. elegans*) | JH3475 | *Smith et al., 2016* | *meg-3* deletion *meg-4* deletion | *meg-3(ax3055)* *meg-4(ax3052)* |
| Antibody | Anti-OLLAS-L2 | Novus Cat# NBP1-06713 | RRID:AB_1625979 | (1:200 IF, 1:1000 Western) |
| Antibody | Anti-PGL-3 KT3 | DSHB Cat# KT3 | RRID:AB_1556927 | (1:10 IF) |
| Antibody | Goat Anti-Mouse IgA 650 | Abcam Cat# ab97014 | RRID:AB_10680780 | (1:200 IF) |
| Antibody | Goat Anti-Rat IgG (H + L) 488 | Thermo Fisher Scientific Cat# A-11006 | RRID:AB_2534074 | (1:200 IF) |
| Antibody | Anti-α-Tubulin | Sigma-Aldrich Cat# T6199 | RRID:AB_477583 | (1:1000 Western) |
| Antibody | Goat Anti-Rat IgG (H + L) HRP | Thermo Fisher Scientific Cat# 31470 | RRID:AB_228356 | (1:2500 Western) |
| Antibody | Goat Anti-Mouse IgG1 HRP | Jackson Immuno Research Labs Cat# 115-035-205 | RRID:AB_2338513 | (1:6000 Western) |
| Sequence-based reagent | dcr12: crRNA to cut *meg-3* at 2408 bp | *Smith et al., 2016* | | tgaaagcttgacagcattcc |

*Continued on next page*

*Continued*

| Reagent type (species) or resource | Designation | Source or reference | Identifiers | Additional information |
|---|---|---|---|---|
| Sequence-based reagent | rHS03: cRNA cuts *meg-3* 5 bp upstream of stop codon | *Smith et al., 2016* | | tcagtacaatcattgatctc |
| Sequence-based reagent | rHS20: crRNA to cut *meg-3* at 2386 bp | This study | | gtcaagctttcagaaatgcg |
| Sequence-based reagent | rHS20: crRNA to cut *meg-3* at 2546 bp | This study | | atccaatcttggaattgtct |
| Sequence-based reagent | rHS26: cRNA to cut MEG-3(HMGL deletion) strain | This study | | tccaatcttggaattgtgcg |
| Sequence-based reagent | rHS01: crRNA to cut *meg-3* at 23 bp | *Smith et al., 2016* | | tcctcaaaaccttacccaag |
| Sequence-based reagent | rHS01: crRNA to cut *meg-3* at 1694 bp | *Smith et al., 2016* | | tcagatcaatcggaacaatg |
| Sequence-based reagent | dcr11: crRNA to cut *meg-4* 3'UTR, 133 bp downstream of stop codon | *Smith et al., 2016* | | tctgcccaggaacttgtaac |
| Sequence-based reagent | pk06: crRNA to cut *meg-4* at 25 bp | *Smith et al., 2016* | | catgtgatctgccaaactcc |
| Sequence-based reagent | dc89: homology template to delete *meg-4* and insert a synthetic guide sequence | *Smith et al., 2016* | | gttgcaggtatgagttctt caaagctttcctcatgtgg gaagtttgtccagagcag aggaacgggtagttttc tattgttatcaggactgctgc |
| Sequence-based reagent | dc257: Homology template to make MEG-3698 | This study | | caccacctcgcatttctga aagcttgacagcattcca atccggattcgccaacg agctcggaccacgtctc atgggaaagtgattgta ccaatttatatctattac ttgtagactata |
| Sequence-based reagent | oHS264: Homology template to delete the HMGL | This study | | ctcaagatccagcttca acctcgccaccacctcg cacaattccaagattg gatggtccttatgccgatgg |
| Sequence-based reagent | oHS270, 272: Homology template to insert MEG-3HMGL-mutations in the HMGL deletion strain | This study | | ctcaagatccagcttcaac ctcgccaccacctcgcat ttctgaaagcttgacagc attttggaggcgcaacag gatgccaacgacgctatt gatactaacgccaaag aaaagacacaactcct gaaagtgaatttggctattc acgggatgtcacctgaaag atggctgtacttgaattattttt gcaccgagacaattccaa gattggatggtccttatgccgatgg |
| Sequence-based reagent | dc198: Homology template to make MEG-3IDR | This study | | gatttttgcaggtatgagctc ctcaaaaccttacccaaa tgtggatgtaaagaga acaccttcctcgtcaatc |

## Worm handling, maternal-effect sterility counts

*C. elegans* was cultured at 20° C according to standard methods (*Brenner, 1974*). To measure maternal-effect sterility, 10 gravid adults were picked to an OP50 plate and allowed to lay eggs

for ~2 hr, then removed. Adult progeny were scored for empty uteri (white sterile phenotype) under a dissecting microscope.

## Identification of MEG-3 HMG-like region

MEG-3 and MEG-4 protein sequences were aligned with HMG boxes from GCNA proteins of *Caenorhabditis* and example vertebrates along with the canonical HMG box of mouse SOX3 using MUSCLE (*Edgar, 2004*). Alignment was manually adjusted according to the published CGNA HMG Hidden Markov Model (*Carmell et al., 2016*). Amino acids were chosen for mutation based on conservation in nematodes.

## CRISPR genome editing

Genome editing was performed in *C. elegans* using CRISPR/Cas9 as described in *Paix et al., 2017*. Strains used in this study, along with guides and repair templates, are listed in *Supplementary file 1*. Some strains were generated in two steps. For example, MEG-3$_{HMGL-}$ was generated by deleting the entire HMGL-like motif in a first step (JH3632) and inserting a modified HMG-like motif with the desired mutations in a second step (JH3861). Genome alterations were confirmed by Sanger sequencing, and expression of tagged strains was verified by immunostaining and western blotting (*Figure 2—figure supplement 1C*).

## Statistical analysis and plotting

On all scatterplots, central bars indicate the mean and error bars indicate one standard deviation. Unless otherwise indicated, differences within three or more groups were evaluated using a one-factor ANOVA and differences between two groups using an unpaired Student's t-test.

## Confocal imaging

Fluorescence confocal microscopy for *Figure 2—figure supplement 1A* and *Figure 6—figure supplements 1*, *2* was performed using a Zeiss Axio Imager with a Yokogawa spinning-disc confocal scanner. Fluorescence confocal microscopy for all other figures was performed using a custom-built inverted Zeiss Axio Observer with CSU-W1 Sora spinning disk scan head (Yokogawa), $1\times/2.8\times$ relay lens (Yokogawa), fast piezo z-drive (Applied Scientific Instrumentation), and a iXon Life 888 EMCCD camera (Andor). Samples were illuminated with 405/488/561/637 nm solid-state laser (Coherent), using a 405/488/561/640 transmitting dichroic (Semrock) and 624-40/692-40/525-30/445-45 nm bandpass filter (Semrock), respectively. Images from either microscope were taken with using Slidebook v6.0 software (Intelligent Imaging Innovations) using a $40\times$–1.3 NA/$63\times$–1.4 NA objective (Zeiss) depending on sample.

## Immunostaining

Adult worms were placed into M9 on poly-l-lysine (0.01%)-coated slides and squashed with a coverslip to extrude embryos. Slides were frozen by laying on aluminum blocks pre-chilled with dry ice for >5 min. Embryos were permeabilized by freeze-cracking (removal of coverslips from slides) followed by incubation in methanol at −20°C for >15 min and in acetone −20°C for 10 min. Slides were blocked in PBS-Tween (0.1%) BSA (0.5%) for 30 min at room temperature and incubated with 50 µL primary antibody overnight at 4°C in a humid chamber. For co-staining experiments, antibodies were applied sequentially (OLLAS before KT3, K76) to avoid cross-reaction. Antibody dilutions (in PBST/BSA): KT3 (1:10, DSHB), K76 (1:10 DSHB), and Rat αOLLAS-L2 (1:200, Novus Biological Littleton, CO), Secondary antibodies were applied for 2 hr at room temperature. Samples were mounted Prolong Diamond Antifade Mountant or VECTASHIELD Antifade Mounting Media with DAPI. Embryos were staged using DAPI stained nuclei and 25 confocal slices spaced 0.18 µm apart and centered on the P cell nucleus were taken using a $63\times$ objective. Unless otherwise indicated, images presented in figures are maximum projections.

## Quantification of immunostaining images

All analyses were performed in ImageJ. For measurements of embryos/cells (*Figure 2B-D*, *3C*), confocal stacks were sum projected and the integrated density was measured within a region of interest. For measurements of condensate intensity and number (*Figure 2B-D*), the 3D objects'

counter function was used with a minimum size of 10 pixels on the full confocal stack confined to a region of interest drawn around the P cell and including objects on edges. The integrated density for all identified particles was summed to give the total intensity in condensates.

## Single-molecule fluorescence in situ hybridization (smFISH)

smFISH probes were designed using Biosearch Technologies's Stellaris Probe Designer, with the fluorophor Quasar670. For sample preparation, embryos were extruded from adults on poly-l-lysine (0.01%) slides and subjected to freeze-crack followed by methanol fixation at >20°C for >15 min. Samples were washed five times in PBS-Tween (0.1%) and fixed in 4% PFA (Electron Microscopy Science, No. 15714) in PBS for 1 hr at room temperature. Samples were again washed four times in PBS-Tween (0.1%), twice in 2× SCC, and once in wash buffer (10% formamide, 2× SCC) before blocking in hybridization buffer (10% formamide, 2× SCC, 200 µg/mL BSA, 2 mM Ribonucleoside Vanadyl Complex, 0.2 mg/mL yeast total RNA, 10% dextran sulfate) for >30 min at 37°C. Hybridization was then conducted by incubating samples with 50 nM probe solutions in hybridization buffer overnight at 37°C in a humid chamber. Following hybridization, samples were washed twice in wash buffer at 37°C, twice in 2× SCC, once in PBS-Tween (0.1%) and twice in PBS. Samples were mounted Prolong Diamond Antifade Mountant.

## Quantification of in situ hybridization images

All measurements were performed on a single confocal slice centered on the P cell nucleus in ImageJ. For early embryos where there is distinct punctate signal (one- and four-cell stage; *Figure 4B, C*, *Figure 4—figure supplements 1*,*2*), a region of interest was drawn, the Analyze Particles feature was used with a manual threshold to identify and measure the integrated density of the puncta. The raw integrated density for all particles in the region of interest was summed to give the total intensity of the mRNA in that region. For 28-cell embryos (*Figure 4D*, *Figure 4—figure supplement 3*), a region of interest was drawn around the $P_4$ blastomere and the intensity of that region was divided by the intensity of a region of the same size in the anterior soma.

## Western blotting of embryonic lysates

Worms were synchronized by bleaching to collect embryos, shaken approximately 20 hr in M9, then plating on large enriched peptone plates with a lawn of *Escherichia coli* NA22 bacteria. Embryos were harvested from young adults (66 hr after starved L1 plating) and sonicated in 2% SDS, 65 mM Tris pH 7, 10% glycerol with protease and phosphatase inhibitors. Lysates were spun at 14,000 rpm for 30 min at 4°C and cleared supernatants were transferred to fresh tubes. Lysates were run on 4–12% Bis-Tris pre-cast gels (Bio-Rad Hercules, CA). Western blot transfer was performed for 1 hr at 4°C onto PVDF membranes. Membranes were blocked overnight and washed in 5% milk, 0.1% Tween-20 in PBS; primary antibodies were incubated overnight at 4°C; secondary antibodies were incubated for 2 hr at room temperature. Membranes were first probed for OLLAS then stripped by incubating in 62.5 mM Tris HCl pH 6.8, 2% SDS, 100 mM ß-mercaptoethanol at 42°C. Membranes were then washed, blocked, and probed for α-tubulin. Antibody dilutions in 5% milk/PBST: Rat α OLLAS-L2 (1:1000, Novus Biological Littleton, CO), Mouse α-tubulin (1:1000, Sigma, St. Louis, MO).

## His-tagged protein expression, purification, and labeling

### Expression and purification of MEG-3 His-tagged fusion proteins

MEG-3 full-length (aa1–862), IDR (aa1–544), Cterm (aa545–862), and HMGL-proteins were fused to an N-terminal 6XHis tag in pET28a and expressed and purified from inclusion bodies using a denaturing protocol (*Lee et al., 2020*). SDS-PAGE gels of purified MEG-3 proteins used in this study are provided in *Figure 5—figure supplement 1*.

Purification of MBP-TEV-PGL-3 was expressed and purified as described (*Putnam et al., 2019*) with the following modifications: MBP was cleaved using homemade TEV protease instead of commercial. A plasmid expressing 8X-His-TEV-8X-Arg tag protease was obtained from Addgene and purified according to the published protocol (*Tropea et al., 2009*). Before loading cleaved PGL-3 protein on to a heparin affinity matrix, cleaved MBP-6X-His and 6X-His-TEV protease were removed using a HisTRAP column (GE Healthcare).

## Protein labeling

Proteins were labeled with succinimidyl ester reactive fluorophores from Molecular Probes (Alexa Fluor 555 or 647 or DyLight 488 NHS Ester) following the manufacturer's instructions. Free fluorophore was eliminated by passage through three Zeba Spin Desalting Columns (7K MWCO, 0.5 mL) into protein storage buffer. The concentration of fluorophore-labeled protein was determined using fluorophore extinction coefficients measured on a Nanodrop ND-1000 spectrophotometer. Labeling reactions resulted in ~0.25–1 label per protein. Aliquots were snap frozen and stored. In phase separation experiments, fluorophore-labeled protein was mixed with unlabeled protein for final reaction concentrations of 25–100 nM of fluorophore-labeled protein.

## In vitro transcription and labeling of RNA

mRNAs were transcribed using T7 mMessageMachine (Thermo Fisher) using the manufacturer's recommendation as described (*Lee et al., 2020*). Fluorescently labeled mRNAs were generated by including a trace amount of ChromaTide Alexa Fluor 488–5-UTP or 546–14-UTP in the transcription reaction. Template DNA for transcription reactions was obtained by PCR amplification from plasmids. Free NTPs and protein were removed by lithium chloride precipitation. RNAs were resuspended in water and stored at −20°C. The integrity of RNA products was verified by agarose gel electrophoresis.

## In vitro condensation experiments and analysis

Protein condensation was induced by diluting proteins out of storage buffer into condensation buffer containing 25 mM HEPES (pH 7.4), salt adjusted to a final concentration of 50 or 150 mM (37.5 mM KCl, 12.5 or 112.5 mM NaCl), and RNA. For MEG-3 and PGL-3 co-condensate experiments with RNA, we used 150 nM MEG-3, 1.8 μM PGL-3. For MEG-3 and PGL-3 co-condensate poly-U(30) competition experiments with *nos-2*, we used 150 nM MEG-3, 2.5 μM PGL-3. For co-assembly experiments in the absence of RNA, we used 150 nM MEG-3, 5 μM PGL-3. MEG-3 and PGL-3 solutions contained 10–50 nM fluorescent trace labels with either 488 or 647 (indicated in the figure legends). MEG-3 and PGL-3 condensation reactions with *nos-2* and poly-U(30) RNA were incubated at room temperature for 30 min before spotting onto a No. 1.5 glass bottom dish (Mattek) and imaged using a 40× with a 1× (used for quantification) and 2.8× (used for display) relay lens oil objective (*Figure 6B*). All other condensate reactions were imaged using thin-chambered glass slides (Erie Scientific Company 30-2066A) with a coverslip (*Figure 6—figure supplements 1*, *2*). Images are single planes acquired using a 40× oil objective over an area spanning 171 × 171 μm.

To quantify the relative intensity of MEG-3 in condensates, a mask was created by thresholding images, filtering out objects of less than four pixels to minimize noise, applying a watershed filter to improve separation of objects close in proximity, and converting to a binary image by the Otsu method using the nucleus counter cookbook plugin. Minimum thresholds were set to the mean intensity of the background signal of the image plus 1–2 standard deviations. The maximum threshold was calculated by adding 3–4 times the standard deviation of the background. Using generated masks, the integrated intensity within each object was calculated. Any normalization is indicated in the corresponding figure legend. Each replicate contained three images each spanning an area of 171 × 171 μm (*Figure 6A*, *Figure 6—figure supplement 1B and C*) or 316.95 × 316.95 μm (*Figure 6C, D*).

## RNA binding by fluorescence filter binding

Proteins were step-dialyzed from 6 M urea into 4.5 M urea, 3 M urea, 1.5 M urea, and 0 M urea in MEG-3 storage buffer (25 mM HEPES, pH 7.4, 1 M NaCl, 6 mM ß-mercaptoethanol, 10% glycerol). RNA-binding reactions consisted of 50 nM 3′ fluorescein-labeled 30U RNA oligonucleotides (poly-U(30)) incubated with either varying protein concentrations (direct binding of polyU(30)) or constant concentrations of protein and varying concentrations of *nos-2* mRNA (long RNA binding by competition) for 30 min at room temperature (final reaction conditions 3.75 mM HEPES, 150 mM NaCl, 0.9 mM, 0.9 mM ß-mercaptoethanol, 1.5% glycerol, 10 mM Tris HCl). Fluorescence filter binding protocol was adapted from a similar protocol using radiolabeled RNA (*Rio, 2012*). Briefly, a pre-wet nitrocellulose was placed on top of Hybond-N+ membrane in a dot-blot apparatus, reactions were applied to the membranes, then washed 2× with 10 mM Tris HCl. Membranes were briefly dried in

air, then imaged using a typhoon FLA-9500 with blue laser at 473 nm. Fraction of RNA bound for each reaction was calculated by dividing the fluorescence signal on the nitrocellulose membrane by the total signal from both membranes.

For direct binding (*Figure 5A*), $K_d$ was calculated by plotting the bound fraction of poly-U(30) RNA as a function of protein concentration and fitting to the following equation in Prism8, where P is the protein concentration in nM, B is the bound fraction of poly-U(30) with non-specific binding subtracted, $B_{max}$ is the maximum specific binding, $K_d$ is the concentration needed to achieve a half-maximum binding at equilibrium, and h is the Hill slope:

$$B = \frac{B_{max} \times P^h}{K_d^h + P^h}$$

For competition binding of *nos-2* RNA (*Figure 5B*), IC50 was calculated by plotting the bound fraction of poly-U(30)as a function of the log of the concentration of *nos-2* and fitting to the following equation, where B is the bound fraction of poly-U(30) in nM, $B_{max}$ is the maximum fraction bound, $B_{min}$ is the minimum fraction bound, X is the concentration of *nos-2* RNA (in nM 30mers), and IC50 is the concentration of *nos-2* RNA (in nM 30mers) needed to achieve half-maximum inhibition of poly-U(30) binding:

$$B = B_{max} + \frac{B_{max} - B_{min}}{1 + 10^{X - log(IC50)}}$$

For a more direct comparison between MEG-3 variants that have a different $K_d$, the EC50 was converted to $K_i$ using the following equation, where [30U] is the concentration of fluorescent labeled poly-U(30) in the reaction in nM and $K_d$ is the experimentally determined $K_d$ for that MEG-3 variant and poly-U(30):

$$K_i = \frac{IC50}{1 + \frac{[30U]}{K_d}}$$

All binding constants are the average value from fitting each replicate separately.

## GST pull-downs

GST fusion proteins were cloned into pGEX6p1 (GE Healthcare, Pittsburgh, PA). MBP fusion proteins were cloned into pJP1.09, a Gateway-compatible pMAL-c2x (*Pellettieri et al., 2003*). Proteins were expressed in Rosetta *E. coli* BL21 cells grown for approximately 4 hr at 37°C, then induced with 1 mM IPTG and grown overnight at 16°C. 200 mg of bacterial pellet of GST fusion proteins was resuspended in 50 mM HEPES, 1 mM EGTA, 1 mM MgCl$_2$, 500 mM KCl, 0.05% NP40, 10% glycerol, pH 7.4 (IP Buffer) with protease and phosphatase inhibitors, lysed by sonication, and bound to magnetic GST beads. Beads were washed and incubated with MBP fusion proteins at 4°C for 1 hr in the same buffer as for lysis. After washing, beads were eluted by boiling and eluates were loaded on SDS-PAGE. Western blot transfer was performed for 1 hr at 4°C onto PVDF membranes. Membranes were blocked and washed in 5% milk, 0.1% Tween-20 in PBS, and incubated with HRP conjugate antibodies. Antibody dilutions in 5% milk/PBST:anti-MBP HRP conjugated, 1:50,000 (NEB, and anti-GST HRP conjugates, 1:2000) (GE Healthcare). Scanned western blot films were quantified using the gel analysis tool in ImageJ.

## Fluorescent protein bead halo assay

Fluorescent protein bead halo assay was adapted from *Patel and Rexach, 2008*. 50 µL of Nickel-NTA agarose beads (Qiagen) were incubated with 50 µL of 10 µM MEG-3 derivatives trace-labeled with Alexa647 or no protein in MEG-3 storage buffer for 1 hr. Beads were washed five times with IP Buffer then blocked for 1 hr in blocking buffer (4 mg/mL BSA, 50 mM HEPES, 1 mM EGTA, 1 mM MgCl$_2$, 500 mM KCl, 0.05% NP40, 10% glycerol, pH 7.4). MBP and PGL-3 trace-labeled with Alexa555 were prepared in PGL-3 storage buffer and diluted to 3 µM in blocking buffer. Additional concentrations of PGL-3 were diluted from this solution to maintain the ratio of label to total protein. 5 µL of blocked MEG-3 or empty beads was added to 50 µL of PGL-3 or MBP solution and incubated for 1 hr. Beads were washed five times in blocking buffer and resuspended in PBS. Beads were

spotted onto a No. 1.5 glass-bottom dish (Mattek) and imaged using a 10× air objective. Images are single planes through the center of the beads.

To quantify the relative intensity of PGL-3 to MEG-3 derivatives on the beads, a mask was created using the Analyze Particles feature in ImageJ on the MEG-3 channel, using a minimum size of $10^{-5}$ cm$^2$. Using the generated mask, the integrated intensity within each bead was calculated for both the MEG-3 and MBP/PGL-3 channels. To remove non-specific binding signal, the mean intensity empty beads incubated with MBP or PGL-3 were subtracted from each pixel yielding the total intensity of each bead. To calculate the intensity ratio for each image, the total intensity on beads of MBP or PGL-3 was divided by the total intensity of MEG-3 on beads. This ratio was normalized to the mean MBP ratio.

## Acknowledgements

We thank the Johns Hopkins Neuroscience Research Multiphoton Imaging Core (NS050274) and the Johns Hopkins Integrated Imaging Center (S10OD023548) for excellent microscopy support. We thank the Page lab for assistance identifying the HMG-like motif, the Nathans lab for the OLLAS antibody, and Addgene for TEV protease from the Waugh lab. We thank Deepika Calidas for collaboration on strains JH3479, JH3517, and JH3420, Mario Martinez for assistance in protein purification, and Baltimore Worm Club and the Seydoux lab for many helpful discussions. This work was supported by the National Institutes of Health (grant number GS: 5R37HD037047, AP: F32GM134630). GS is an investigator of the Howard Hughes Medical Institute.

## Additional information

### Competing interests

Geraldine Seydoux: serves on the Scientific Advisory Board of Dewpoint Therapeutics, Inc. The other authors declare that no competing interests exist.

### Funding

| Funder | Grant reference number | Author |
| --- | --- | --- |
| National Institutes of Health | 5R37HD037047 | Helen Schmidt<br>Andrea Putnam<br>Geraldine Seydoux |
| National Institutes of Health | F32GM134630 | Andrea Putnam |
| Howard Hughes Medical Institute | | Geraldine Seydoux<br>Dominique Rasoloson |

The funders had no role in study design, data collection and interpretation, or the decision to submit the work for publication.

### Author contributions

Helen Schmidt, Andrea Putnam, Conceptualization, Data curation, Formal analysis, Validation, Investigation, Visualization, Methodology, Writing - original draft, Writing - review and editing; Dominique Rasoloson, Resources, Validation, Investigation, Writing - original draft, Writing - review and editing; Geraldine Seydoux, Conceptualization, Supervision, Funding acquisition, Writing - original draft, Writing - review and editing

### Author ORCIDs

Helen Schmidt https://orcid.org/0000-0002-3449-2790
Andrea Putnam https://orcid.org/0000-0001-7985-142X
Dominique Rasoloson http://orcid.org/0000-0003-2210-1569
Geraldine Seydoux https://orcid.org/0000-0001-8257-0493

Decision letter and Author response
Decision letter https://doi.org/10.7554/eLife.63698.sa1
Author response https://doi.org/10.7554/eLife.63698.sa2

## Additional files

### Supplementary files

• Supplementary file 1. *C. elegans* strains used in this study, generated by CRISPR/Cas9 genome editing.

• Transparent reporting form

### Data availability

All data generated or analyzed during this study are included in the manuscript and supporting files. Source data files have been provided for Figures 2–7.

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
