## [Decision Letter]

**Acceptance summary:**

Previous work from the Seydoux lab showed that P granules are multi-layered, with a liquid-like internal compartment formed by PGL proteins and gel like outer compartments formed by the RNA binding proteins, MEG-3/4, that assemble on the surface of the PGL compartment. Here they have used a combination of in vitro reconstitution and in vivo experiments to dissect MEG-3, to understand how interactions of its different regions produce this organization. They find that a C-terminal MEG-3 fragment is the dominant element necessary to form PGL-containing condensates in vivo. An HMGL motif in the C-terminus appears to contribute to both functions. In contrast, the N-terminal IDR of MEG-3 binds RNA in vitro, but does not form robust condensates in vivo. This leads to a model in which MEG-3 is recruited to PGL condensates through its C-terminal region, and all elements of the protein are necessary to produce fully functional P granules.

**Decision letter after peer review:**

Thank you for submitting your article "Coordination of RNA and protein condensation by the P granule protein MEG-3" for consideration by *eLife*. Your article has been reviewed by 2 peer reviewers, and the evaluation has been overseen by a Reviewing Editor and James Manley as the Senior Editor. The reviewers have opted to remain anonymous.

The reviewers have discussed the reviews with one another and the Reviewing Editor has drafted this decision to help you prepare a revised submission.

Summary:

Previous work from the Seydoux lab showed that P granules are multi-layered, with a liquid-like internal compartment formed by PGL proteins and gel like outer compartments formed by the RNA binding proteins, MEG-3/4, that assemble on the surface of the PGL compartment. Here they have used a combination of in vitro reconstitution and in vivo experiments to dissect MEG-3, to understand how interactions of its different regions produce this organization.

They find that a C-terminal MEG-3 fragment is the dominant element necessary to form PGL-containing condensates in vivo. An HMGL motif in the C-terminus appears to contribute to both functions. In contrast, the N-terminal IDR of MEG-3 binds RNA in vitro, but does not form robust condensates in vivo. This leads to a model in which MEG-3 is recruited to PGL condensates through its C-terminal region, and all elements of the protein are necessary to produce fully functional P granules.

The reviewers feel that this represents nice work. However both have some issues with the determination of the binding site of the Meg-3 protein. In addition, both reviewers feel that the correlation between the in vitro and in vivo data is not very good.

Essential revisions:

There are several essential revisions the reviewers feel should be done, pertaining to Figure 2. First, they would like it to be shown that RNA is co-recruited into the granule and to determine the region in the C terminus that is necessary for localization and second, whether the IDR is necessary for PGL binding.

Title: Somewhat overstates the data since the coordination of RNA and protein is not fully characterized.

The full reviews are included below.

*Reviewer #1:*

Previous work from the Seydoux lab showed that P granules are multi-layered, with a liquid-like internal compartment formed by PGL proteins and gel like outer compartments formed by the RNA binding proteins, MEG-3/4, that assemble on the surface of the PGL compartment. Here they have used a combination of in vitro reconstitution and in vivo experiments to dissect MEG-3, to understand how interactions of its different regions produce this organization.

They find that a C-terminal MEG-3 fragment binds PGL-3 in vitro and is the dominant element necessary to form PGL-containing condensates in vivo. An HMGL motif in the C-terminus appears to contribute to both functions. In contrast, the N-terminal IDR of MEG-3 binds RNA in vitro, but does not form robust condensates in vivo. Both fragments of MEG-3, including the HMGL motif, are needed to generate RNA foci in vivo. This leads to a model in which MEG-3 is recruited to PGL condensates through its C-terminal region, and all elements of the protein are necessary to produce fully functional P granules.

I have two main concerns regarding the data and story:

1. As an *eLife* paper on the molecular mechanism by which MEG-3 acts to create P granules, the work does not go far enough in understanding how the sequence and structural elements of MEG-3 interact with PGL condensates and RNA to produce the full granule.

It is unclear how we should think about the functionality of the C-terminal fragment of MEG-3. Is the whole element a single folded domain, which mediates binding to PGL-3, layered condensate formation, and localization of PGL condensates to one pole of the embryo? Or is the fragment a folded HMGL domain surrounded by either other domains or partially disordered elements with independent activities? The hydrophobicity plot in Figure 1 suggests a folded domain might exist at ~residues 720-880. Relatedly, does mutating the HMGL motif disrupt a binding surface, or unfold a domain? What region of the MEG-3 C-terminus mediates localization in vivo? Apparently not the HMGL motif, as the HMGL- mutant still localizes properly. Without such information it is hard to precisely understand the activities of the C-terminal fragment.

Further, the authors should make an effort to understand why the MEG-3_IDR_ forms homogeneous condensates with PGL-3, while the C-terminal fragment and HMGL- protein remain demixed, forming a heterogeneous structure. What sequence features or binding properties are necessary for demixing? This multi-layered architecture is an important feature of P granules and should be addressed.

Finally, why does the MEG-3_IDR_ bind RNA with ~7-fold higher affinity than the full-length protein? This suggests some autoinhibition in the protein. Can the C-terminus act in trans to decrease the affinity of the IDR for RNA? If so, does it bind to the IDR?

2. The in vitro and in vivo data do not correlate particularly well, clouding a clear mechanistic picture of the cellular behaviors. The authors acknowledge this issue in some areas, but we are still left not understanding substantial differences between the molecular behaviors in the two settings. For example, it is not clear how to reconcile the co-condensation with PGL-3 of the MEG-3_IDR_ and HMGL- proteins in vitro with the cellular behaviors of these proteins. From the data presented, both robustly phase separate with PGL-3 (figure 2), but do not co-assemble in vivo (figure 4). This does not seem to be a matter of degree, as the HMGL- construct assembles with PGL-3 in vitro identically to WT MEG-3. Quantifying the effect of the HMGL- mutations on affinity for PGL-3 in physiologic conditions may clarify.

It is similarly unclear why the MEG-3_IDR_ and HMGL- proteins form foci in cells, and should be able to bind RNA with high affinity (the former better than WT MEG-3) based on in vitro data, do not recruit Y51F10.2 into those foci. Even if RNA levels are reduced in embryos expressing the mutants, one would expect the RNA that is expressed to be recruited into the protein foci. Along these lines, in figure 2, the authors should examine co-recruitment of RNA into the MEG-3/PGL-3 condensates. Recruitment should track with binding, but it is important to show this, in part to compare with the cellular results in Figure 5. Perhaps such experiments would show that there is no in vitro/in vivo discrepancy here, although in that case one would have to explain why IDR and HMGL- can bind RNA but do not recruit it into their condensates.

Two technical points relate to these concerns:

First, the authors should quantify the binding data in Figure 2, showing the interactions of PGL proteins with the MEG-3 C-terminus, ideally determining the Kd of the interactions. They claim the interaction is of high affinity, but this cannot be inferred from a single blot performed at a single concentration. Similarly, the magnitude of the change in affinity induced by mutation of the HMGL motif cannot be inferred from these data, especially given that the amount of GST fusion was higher in the HMGL- protein than in the WT C-term protein (cf. GST bands at bottom of blot).

Second, it is disappointing that the MEG-3_IDR_ was not analyzed for direct binding to PGL proteins. I understand that the GST-fusion did not express, but there are a variety of ways around this technical difficulty, especially given that the His-tagged version of the protein could be expressed and purified. The authors imply that the dominant interaction is mediated by the C-terminus, but the co-LLPS of the IDR with PGL-3 indicates the IDR can also interact with some affinity.

*Reviewer #2:*

The manuscript by Schmidt et al. performs a functional dissection of the *C. elegans* MEG-3 protein in assembling P granules. P granules are a prime model for studying the properties of ribonucleoprotein condensates but how condensates like P granules are actually assembled is less well understood. Previous work has shown that MEG-3, along with the redundantly functioning MEG-4, is required to localize PGL-1 and PGL-3 condensates to the posterior of the embryo. MEG-3/4 form a gel like shell that surrounds a more liquid-like core comprising PGL-1 and PGL-3. In addition, MEG-3 binds to RNA and is thought to recruit RNAs to P granules. How MEG-3 can perform these functions to assemble P granules is the focus of this paper. The authors perform both in vitro and in vivo experiments testing the activities of different domains/regions of the MEG-3 protein. They find that the HMG domain in the C-terminal half of MEG-3 is required for MEG-3 to interact with PGL-3 and that the C-terminal half (including the HMG domain) are required for PGL-3 condensates to localize in the posterior of the embryo, whereas the IDR in the N-terminal half of MEG-3 binds to RNA and recruits RNA to P granules. Notably, the IDR and RNA recruitment are not required for MEG-3 condensation, its localization, or its ability to bind to, localize, and stabilize PGL-3 condensates. The authors guardedly conclude that "germ granule assembly depends at least in part on protein-protein interactions that drive protein condensation independent of RNA".

Overall, this is really nice work – the experiments are carefully executed, the data are convincing, and the conclusions are solid. I have two main concerns:

1) From the data presented, it seems likely that the functions the authors map to MEG-3_Cterm_ can be attributed to the HMG domain but the authors carefully evade this issue. It seems like this is an important distinction to make and would provide significantly more resolution to the analysis. Testing the function of MEG-3_Cterm_ with the HGML mutation would help to answer the question.

2) The authors seem overly timid in making connections between their findings and in their conclusions. For example, it seems like they want to argue that RNA is largely a passenger rather than an instigator in P granule assembly but are afraid to make that case. In addition, the discussion would benefit from a more thorough fleshing out of their model, with better explanation of how they envision the different domains are MEG-3 are regulating localization, condensation, stabilization of PGL condensates, recruitment of RNA, etc. Why does MEG-3_Cterm_ only condense in the posterior even though it is not localized, why are all the domains are needed for RNA recruitment?

*Reviewer #3:*

Previously, the Seydoux lab discovered a class of genes, the meg genes, which encode for intrinsically disordered phospho-regulated proteins, and which are required for P granule segregation in *C. elegans* embryos (Wang et al., 2014). In further studies the lab showed that the MEG-3 protein can bind to RNA and that MEG-3 can phase separate in vitro in the presence of RNA. Importantly, the authors find that the intrinsically disordered region and the RNA concentrations tune phase separation (Smith et al., 2016). In further work, the lab suggested, that MEG-3 is in a rather gel-like protein phase which does not dissolve upon temperature shifts in vivo (Putnam et al., 2019) and which has different material properties than the PGL phase which is liquid-like and forms by phase separation. Both, the MEG phase and the PGL phase overlap but do not seem to mix, both in vitro and in vivo, and the question is how this process could be regulated.

In a recent study, the lab looked for interacting RNAs in *C. elegans* embryos by using crosslinking approaches in combination with immunoprecipitations. Surprisingly, they found the majority of mRNA’s bound by MEG-3 after immunoprecipitations rather than to the RGG domain P granule protein PGL-1. Several of the identified mRNA’s indeed colocalize with MEG-3 in combined immunofluorescence/FISH experiments and show as well P lineage enrichment. In the same study, in vitro assays with MEG-3 and RNA indicated that the MEG-3 protein aggregates in the presence of low RNA concentrations and high or low salt conditions, whereas it associates into condensates with high RNA concentrations and medium salt concentrations.

Now, in the new study by Schmidt et al. the authors further study the protein MEG-3 and do an in vivo and in vitro structure/function analysis of the protein. They suggest that MEG-3 coordinates RNA and protein condensation, propose that the C-terminus is “necessary and sufficient to build MEG-3/PGL co-condensates independent of RNA” (line 16 of the abstract) and suggest that the IDR is required but not sufficient for RNA recruitment to P granules. This is a good paper and has lots of excellent data on the role of the MEGs.

However, the current manuscript has several inconsistencies with their previous work.

Here are the detailed arguments:

MEG-3 seems to contain a C-terminal HMG-like motif, which seems to interact in vitro with the PGL P granule proteins.

In vitro, the C-terminal part of MEG-3 seems to interact with PGL-1 and PGL-3 in pull down assays. Mutations of conserved residues in the HMG-like motif strongly reduces the PGL binding. But how does full length MEG-3 behave in the same pull down assay? Do the mutations in the HMG-like motif as well strongly reduce the interaction? How do the authors conclude that the HMG domain is required for “high affinity” binding to PGL proteins? (see line 162). Whether these interactions play a role in vivo remains unclear as well, especially as MEGs and PGL proteins seem to have very different biophysical material properties.

MEG-3 interacts with RNA by the IDR region only.

These results are inconsistent with previous experiments. Full length MEG-3 interacts with polyU RNA in the low nM (around 30nM) range, whereas the IDR region shows around 15 fold lower binding affinities (around 500nM) (see Smith et al., 2016). Now Schmidt et al. report that full length MEG-3 binds to nos-2 RNA in the high nM range (around 800nM, roughly 30 fold lower), whereas the IDR region binds in the low nM range (around 100nM, which is around 8 fold higher). These differences may be explained by the differences in RNA, but could as well represent differences in the MEG-3 protein quality and differences in the assay. MEG-3 protein as well as its fragments tend to aggregate and do not seem spherical as published previously (compare Fig. 2A from this manuscript with Fig. 4 and Fig 4S from Lee et al., 2020). Protein quality control as well as performing gel shift assays, as in previous report, would indicate the quality of the protein and be important to resolve these inconsistencies. Partially aggregated MEG-3 might be present in the filter binding assays and eventually lead to much lower apparent RNA binding constants for the MEG-3 full length protein.

In further in vitro assays it is tested whether MEG-3 interacts with PGL-3 condensates. Indeed, all MEG-3 fragments interact with the PGL condensates and form domains on the PGL condensates in the presence of RNA (see Fig. 2C). In the absence of RNA the MEG-3 IDR domain fully mixes with the PGL domain indicating strong interaction with PGL-3. However, this domain was not tested for interaction with PGL-3 by any other assays, and in vivo does not interact with PGLs.

Co-assembly of MEG/PGL condensates is driven by the MEG-3 C-terminus.

The MEG-3 fragments tested in vitro are now generated in vivo by replacing the endogenous meg-3 locus with the gene variants by CRISPR/Cas9. The authors do a series of immunofluorescence experiments in the meg-3 variant lines and stain for MEG-3 and PGL-3. The authors state that “in embryos expressing MEG-3 Cterm, PGL-3 condensates localized properly in P1…”. However, as their images (see Fig4A) as well as their quantification indicates, this is not the case. Only full length MEG-3 shows proper localization of PGL-3, and neither the C-terminal, nor the IDR or the HMGL variant show correct PGL-3 localisation. Although there seems to be an enrichment of PGL-3 in P1 in the Cterm mutant, clearly other parts of the protein contribute to PGL-3 localization as seen in MEG-3 full length (see Fig. 4C)

Efficient recruitment of mRNA to P granules requires all parts of the MEG-3 protein. Previously, the Seydoux lab discovered that hundreds of mRNA crosslink to MEG-3 in vivo and several of these mRNA are indeed enriched in the P lineage (Lee et al., 2020). Now, they test which part of MEG-3 is required for enrichment of Y51F10.2. Intriguingly, Y51F10.2 becomes only considerably enriched in P4 cells if all domains of the MEG-3 protein get expressed, indicating that enrichment of PGL’s by the MEG-3Cterm construct is not sufficient for enrichment of Y51F10.2 mRNA. However, to address if mRNA recruitment requires indeed all MEG-3 domains it would be much better to quantify the level of polyA enrichment in the different MEG-3 mutants (see Fig 5 supp1). Previously, Lee et al have shown that nearly all polyA RNA which is accessible to FISH in early embryos is located in the P lineage in the P2 cell and this depends on the presence of wildtype copies of meg-3 and meg-4 (see Fig 1F, Lee et al., 2020). And if indeed the IDR of MEG-3 recruits mRNA than the MEG-3Cterm mutant should not enrich any mRNA in P1 to P4. Importantly, very little polyA RNA is enriched in pgl-1 pgl-3 mutants indicating that MEG-3 enrichment is not sufficient for P lineage enrichment of mRNA.

---

## [Author Response]

Essential revisions:There are several essential revisions the reviewers feel should be done, pertaining to Figure 2. First, they would like it to be shown that RNA is co-recruited into the granule and to determine the region in the C terminus that is necessary for localization and second, whether the IDR is necessary for PGL binding.

Thank you for these recommendations. To address the reviewers’ comments, we have reworked our entire in vitro analyses examining MEG-3 binding to RNA and PGL-3 and MEG-3 condensation with and without PGL-3. The new data address what is required for RNA to be recruited to MEG/PGL co-condensates (robust condensation driven by both the IDR and Cterm), and what MEG-3 domains contribute to PGL binding (the HMGL domain and NOT the IDR). The new data are presented in three new figures (Figures 5, 6 and 7) that replace Figure 2 in the original submission.

The revised in vitro data better aligns with the in vivo data and supports the main thesis of the paper: MEG-3 is a modular protein with separate domains required for RNA binding (IDR, aa 1-544), PGL-3 binding (HMGL domain, aa 700-740) and protein condensation (aa 544 to 862). The three domains synergize in vivo to assemble P granules: MEG-3/PGL-3 co-condensates that recruit and protect from degradation specific maternal RNAs.

Title: Somewhat overstates the data since the coordination of RNA and protein is not fully characterized.

We have changed the title and abstract to better reflect the broader take-home message of the paper: protein condensation mechanisms drive P granule assembly. As described in the Introduction and Discussion, recent studies examining stress granules have highlighted the role of RNA in RNA granule assembly. Our findings illustrate a different paradigm where RNA plays a more passive role and protein-protein interactions drive condensate assembly.

The full reviews are included below.Reviewer #1:[…] I have two main concerns regarding the data and story:1. As an eLife paper on the molecular mechanism by which MEG-3 acts to create P granules, the work does not go far enough in understanding how the sequence and structural elements of MEG-3 interact with PGL condensates and RNA to produce the full granule.It is unclear how we should think about the functionality of the C-terminal fragment of MEG-3. Is the whole element a single folded domain, which mediates binding to PGL-3, layered condensate formation, and localization of PGL condensates to one pole of the embryo? Or is the fragment a folded HMGL domain surrounded by either other domains or partially disordered elements with independent activities? The hydrophobicity plot in Figure 1 suggests a folded domain might exist at ~residues 720-880. Relatedly, does mutating the HMGL motif disrupt a binding surface, or unfold a domain? What region of the MEG-3 C-terminus mediates localization in vivo? Apparently not the HMGL motif, as the HMGL- mutant still localizes properly. Without such information it is hard to precisely understand the activities of the C-terminal fragment.

We have reworked our in vitro analysis of MEG-3 variants. By comparing proteins with and without mutations in the HMGL domain in the context of full-length MEG-3 and MEG-3_C-term_, we have found that the HMGL domain is only required for binding to PGL-3 and is dispensable for condensation and localization in vitro.

These observations indicate that the C-terminal domain of MEG-3 codes for two independent activities: binding to PGL-3 driven by the HMGL motif (aa 700-740) and condensation in posterior cytoplasm driven by aa 740 to 862, and possibly sequences N-terminal to the HMGL motif (598-698) – although these are not sufficient for condensation in the context of MEG-3_698_. As the reviewer suggests, our observations are consistent with mutations in the HMGL motif affecting only a PGL-3 binding surface and not the entire C-terminal domain, since MEG-3_Cterm_ and MEG-3_C-termHMGL-_ exhibited identical condensation properties in vitro. We do not know yet whether the C-terminal sequences responsible for condensation fold into a globular domain or not. (MEG-3_C-termHMGL-_ was not expressed at high enough levels to be analyzed in vivo).

Further, the authors should make an effort to understand why the MEG-3_IDR_ forms homogeneous condensates with PGL-3, while the C-terminal fragment and HMGL- protein remain demixed, forming a heterogeneous structure. What sequence features or binding properties are necessary for demixing? This multi-layered architecture is an important feature of P granules and should be addressed.

Our new findings indicate that de-mixing correlates with conditions that favor robust MEG-3 condensation. Full length MEG-3 condense most efficiently, followed by MEG-3_C-term_ and the MEG-3_IDR_ which condenses poorly at concentrations in the physiological range (150 nM; Figure 6A). In the presence of high RNA, which has a strong solubilizing influence on MEG-3 (Lee et al., 2020), only full-length MEG-3 fully de-mixes from PGL-3 (Figure 6B). Under low or no RNA conditions, both full-length MEG-3 and MEG-3_C-term_ de-mix (Figure 6 Supplement 3). The MEG-3_IDR_ does not de-mix under any conditions (Figure 6 and supplement). The HMGL domain does not contribute to MEG-3 condensation in vitro and is not required for de-mixing under any condition.

Finally, why does the MEG-3_IDR_ bind RNA with ~7-fold higher affinity than the full-length protein? This suggests some autoinhibition in the protein. Can the C-terminus act in trans to decrease the affinity of the IDR for RNA? If so, does it bind to the IDR?

The difference in RNA binding between full length MEG-3 and MEG-3_IDR_ was observed using the non-physiological RNA poly-U30. To examine whether this difference holds when using a physiological substrate, we examined binding to *nos-2* RNA using a competition assay (Figure 5 B). We detected *no difference* between full length MEG-3 and MEG-3_IDR_ (Figure 5 B) in binding to *nos-2* RNA.

We did test the possibility that the C-terminus acts in trans to reduce the affinity of the IDR for poly-U as suggested by the reviewer. These experiments, however, were confounded by the fact that the C-terminus also binds poly-U (albeit with much lower affinity) complicating the assay, which required using high concentrations of the C-terminus. We therefore did not include these experiments in the manuscript but present them as Author response images 1-3.

MEG-3_Cterm_ RNA binding competition assays:

In these experiments, we examined binding of MEG-3_IDR_ (105nM) to polyU RNA (50nM) in the presence of increasing concentrations of MEG-3_Cterm_.

Replication of a normally *intramolecular* inhibitory interaction *in trans* is expected to require high concentration of the inhibitory domain. We therefore first tested whether how much RNA is bound by MEG-3_Cterm_ at high concentrations. We found that at micromolar concentration, MEG-3_Cterm_ binding to RNA becomes non-negligible, binding as much as 15% of RNA.

When MEG-3_IDR_ and MEG-3_Cterm_ were pre-mixed and added to poly-U30 and allowed to equilibrate in binding buffer solution, we observed that the fraction of bound RNA remained roughly constant with increasing MEG-3_Cterm_.

**Author response image 2. respfig2:** 

Subtracting the fraction of RNA bound by the MEG-3_Cterm_ (observed when MEG-3 is tested alone) from the fraction of RNA bound by the combination of the MEG-3_IDR_ and MEG-3_Cterm_, we observed a decrease in bound RNA with increasing MEG-3_Cterm_.

**Author response image 3. respfig3:** 

These results indicate that mixing of the MEG-3_IDR_ and MEG-3_Cterm_ results in lower RNA binding than would be expected from the sum of their independent binding, consistent with competition. However, because both domains bind RNA under these conditions, we were not able to determine conclusively whether the MEG-3_Cterm_ inhibits MEG-3_IDR_ binding to RNA or *vice versa*. We prefer therefore not to draw any conclusions from these experiments.

2. The in vitro and in vivo data do not correlate particularly well, clouding a clear mechanistic picture of the cellular behaviors. The authors acknowledge this issue in some areas, but we are still left not understanding substantial differences between the molecular behaviors in the two settings. For example, it is not clear how to reconcile the co-condensation with PGL-3 of the MEG-3_IDR_ and HMGL- proteins in vitro with the cellular behaviors of these proteins. From the data presented, both robustly phase separate with PGL-3 (figure 2), but do not co-assemble in vivo (figure 4). This does not seem to be a matter of degree, as the HMGL- construct assembles with PGL-3 in vitro identically to WT MEG-3. Quantifying the effect of the HMGL- mutations on affinity for PGL-3 in physiologic conditions may clarify.It is similarly unclear why the MEG-3_IDR_ and HMGL- proteins form foci in cells, and should be able to bind RNA with high affinity (the former better than WT MEG-3) based on in vitro data, do not recruit Y51F10.2 into those foci. Even if RNA levels are reduced in embryos expressing the mutants, one would expect the RNA that is expressed to be recruited into the protein foci. Along these lines, in figure 2, the authors should examine co-recruitment of RNA into the MEG-3/PGL-3 condensates. Recruitment should track with binding, but it is important to show this, in part to compare with the cellular results in Figure 5. Perhaps such experiments would show that there is no in vitro/in vivo discrepancy here, although in that case one would have to explain why IDR and HMGL- can bind RNA but do not recruit it into their condensates.

We have identified in vitro conditions that best correlate with in vivo observations. We now present the in vitro data AFTER the in vivo data to make it clear that we evaluated several in vitro conditions (Figure 6 Supplement 1) and identified those that most closely reproduced the in vivo observations (150mM NaCl, 150nM MEG-3 and 20ng/ul RNA, all within the range of estimated physiological conditions). Under these conditions, we find that full length MEG-3 is required to promote maximum MEG-3 condensation and RNA recruitment as is observed in vivo. The C-term performs better than the IDR in promoting condensation, but is less efficient than full length MEG-3, and does not recruit RNA as efficiently as either full length MEG-3 or the IDR (Figure 6A).

One difference that remains between in vitro and in vivo findings is that the IDR and MEG-3_HMGL-_ are recruited to PGL-3 condensates in vitro but not in vivo.

To address this discrepancy, we built on our prior finding that MEG-3 binds to PGL-3 directly in a GST-pull down assay. We developed a second binding assay that allowed us to examine the MEG-3_IDR_, MEG-3_Cterm_ and MEG-3_CtermHMGL-_ in parallel for binding to PGL-3. In this assay, we found that only wild-type MEG-3_Cterm_ binds to PGL-3. (The binding assays are done under conditions where PGL-3 does not form condensates).

These results suggest that co-assembly of MEG-3 and PGL-3 condensates in vivo depends on a direct binding interaction between MEG-3 and PGL-3 mediated by the HMGL domain. We hypothesize that the complexity of the cytoplasm makes for a competitive environment not reproduced in our in vitro reconstitutions with purified proteins. Consequently, specific binding between MEG-3 and PGL-3 mediated by the HMGL is required for co-assembly in vivo but not in vitro.

We showed previously that association with PGL-3 condensates stimulates MEG-3 condensation (Putnam et al. 2019). As expected therefore, the condensation efficiency of MEG-3_HMGL-_ was reduced compared to wild-type in vivo (Figure 2 D). We suggest that this reduced condensation efficiency may also account for why MEG-3_HMGL-_ fails to assemble support RNA recruitment in vivo (Figure 4).

In conclusion, although the new in vitro conditions do not fully recapitulate the complex environment of the *C. elegans* cytoplasm, they have allowed us to define three regions in MEG-3 (IDR, HMGL motif, and C-terminal domain) with distinct activities that play complementary roles in P granule assembly. The take-home message overall is that protein-protein and condensate-condensate interactions are the primary drivers of P granule assembly. We have rewritten the abstract and title to better reflect this emphasis.

Two technical points relate to these concerns:First, the authors should quantify the binding data in Figure 2, showing the interactions of PGL proteins with the MEG-3 C-terminus, ideally determining the Kd of the interactions. They claim the interaction is of high affinity, but this cannot be inferred from a single blot performed at a single concentration. Similarly, the magnitude of the change in affinity induced by mutation of the HMGL motif cannot be inferred from these data, especially given that the amount of GST fusion was higher in the HMGL- protein than in the WT C-term protein (cf. GST bands at bottom of blot).

We have addressed these concerns in a new bead-based assay for protein-protein interactions (Figure 7 C). The new assay utilizes purified proteins, which allowed us to compare different concentrations of MEG-3_Cterm._

Second, it is disappointing that the MEG-3_IDR_ was not analyzed for direct binding to PGL proteins. I understand that the GST-fusion did not express, but there are a variety of ways around this technical difficulty, especially given that the His-tagged version of the protein could be expressed and purified. The authors imply that the dominant interaction is mediated by the C-terminus, but the co-LLPS of the IDR with PGL-3 indicates the IDR can also interact with some affinity.

We have now tested the MEG-3_IDR_ for PGL-3 binding in the new bead-based assay and found that the MEG-3_IDR_ does NOT bind PGL-3, unlike MEG-3_Cterm_.

Reviewer #2:[…] Overall, this is really nice work – the experiments are carefully executed, the data are convincing, and the conclusions are solid. I have two main concerns:1) From the data presented, it seems likely that the functions the authors map to MEG-3_Cterm_ can be attributed to the HMG domain but the authors carefully evade this issue. It seems like this is an important distinction to make and would provide significantly more resolution to the analysis. Testing the function of MEG-3_Cterm_ with the HGML mutation would help to answer the question.

As suggested, we have tested the properties MEG-3_Cterm_ with the HGML mutation (Figures 6-7). We find that the HMGL- mutation does not affect condensation and recruitment in vitro, but severely reduces binding to PGL-3. These findings confirm that the C-terminus has two independent activities: driving MEG-3 condensation which does NOT require the HMGL motif, and binding to PGL-3 which requires HMGL.

The HMGL domain is essential for assembly of PGL-MEG co-condensates in vivo but not in vitro. We speculate this discrepancy is due to the presence of other factors or condensates in vivo that compete with PGL-3 condensates for co-assembly with MEG-3.

We previously showed that PGL-3 stimulates MEG-3 condensation (Putnam et al., 2019). The observation that MEG-3_HMGL-_ does not condense as efficiently as wild-type in vivo, therefore, is consistent with the inability of MEG-3_HMGL-_ to interact efficiently with PGL-3 in vivo. We have clarified these points in the text.

2) The authors seem overly timid in making connections between their findings and in their conclusions. For example, it seems like they want to argue that RNA is largely a passenger rather than an instigator in P granule assembly but are afraid to make that case. In addition, the discussion would benefit from a more thorough fleshing out of their model, with better explanation of how they envision the different domains are MEG-3 are regulating localization, condensation, stabilization of PGL condensates, recruitment of RNA, etc. Why does MEG-3_Cterm_ only condense in the posterior even though it is not localized, why are all the domains are needed for RNA recruitment?

We agree and have expanded the discussion to make these points more clearly and reworked the title and abstract to emphasize our main finding: that P granule assembly is driven primarily by protein-based condensation mechanisms.

Reviewer #3:Previously, the Seydoux lab discovered a class of genes, the meg genes, which encode for intrinsically disordered phospho-regulated proteins, and which are required for P granule segregation in *C. elegans* embryos (Wang et al., 2014). In further studies the lab showed that the MEG-3 protein can bind to RNA and that MEG-3 can phase separate in vitro in the presence of RNA. Importantly, the authors find that the intrinsically disordered region and the RNA concentrations tune phase separation (Smith et al., 2016). In further work, the lab suggested, that MEG-3 is in a rather gel-like protein phase which does not dissolve upon temperature shifts in vivo (Putnam et al., 2019) and which has different material properties than the PGL phase which is liquid-like and forms by phase separation. Both, the MEG phase and the PGL phase overlap but do not seem to mix, both in vitro and in vivo, and the question is how this process could be regulated.In a recent study, the lab looked for interacting RNAs in *C. elegans* embryos by using crosslinking approaches in combination with immunoprecipitations. Surprisingly, they found the majority of mRNA’s bound by MEG-3 after immunoprecipitations rather than to the RGG domain P granule protein PGL-1. Several of the identified mRNA’s indeed colocalize with MEG-3 in combined immunofluorescence/FISH experiments and show as well P lineage enrichment. In the same study, in vitro assays with MEG-3 and RNA indicated that the MEG-3 protein aggregates in the presence of low RNA concentrations and high or low salt conditions, whereas it associates into condensates with high RNA concentrations and medium salt concentrations.Now, in the new study by Schmidt et al. the authors further study the protein MEG-3 and do an in vivo and in vitro structure/function analysis of the protein. They suggest that MEG-3 coordinates RNA and protein condensation, propose that the C-terminus is “necessary and sufficient to build MEG-3/PGL co-condensates independent of RNA” (line 16 of the abstract) and suggest that the IDR is required but not sufficient for RNA recruitment to P granules. This is a good paper and has lots of excellent data on the role of the MEGs.However, the current manuscript has several inconsistencies with their previous work.Here are the detailed arguments:MEG-3 seems to contain a C-terminal HMG-like motif, which seems to interact in vitro with the PGL P granule proteins.In vitro, the C-terminal part of MEG-3 seems to interact with PGL-1 and PGL-3 in pull down assays. Mutations of conserved residues in the HMG-like motif strongly reduces the PGL binding. But how does full length MEG-3 behave in the same pull down assay? Do the mutations in the HMG-like motif as well strongly reduce the interaction? How do the authors conclude that the HMG domain is required for “high affinity” binding to PGL proteins? (see line 162). Whether these interactions play a role in vivo remains unclear as well, especially as MEGs and PGL proteins seem to have very different biophysical material properties.

We have expanded our analyses of the MEG-3/PGL-3 interaction using a new bead halo assay using purified proteins (Figure 7C). The new data confirms that MEG-3 interacts with PGL-3 via the HMGL motif. Unlike the C-term, the IDR does not interact with PGL-3. Additionally we have modified our language and no longer characterize this interaction as “high affinity” since our data do not address binding affinity, only specificity.

MEG-3 interacts with RNA by the IDR region only.These results are inconsistent with previous experiments. Full length MEG-3 interacts with polyU RNA in the low nM (around 30nM) range, whereas the IDR region shows around 15 fold lower binding affinities (around 500nM) (see Smith et al., 2016). Now Schmidt et al. report that full length MEG-3 binds to nos-2 RNA in the high nM range (around 800nM, roughly 30 fold lower), whereas the IDR region binds in the low nM range (around 100nM, which is around 8 fold higher). These differences may be explained by the differences in RNA, but could as well represent differences in the MEG-3 protein quality and differences in the assay.

The differences are likely due to 1) the new assay used to measure RNA binding and 2) improvements in our protein purification protocols.

The assay used previously was based on fluorescence anisotropy. Since phase separation can alter the fluorescence anisotropy signal, this assay suffers potentially from complications resulting from phase separation that occurs at different concentrations for MEG-3 and variants.

We also have added a size exclusion step to the protein purification protocol described in Smith et al 2016 which significantly improves protein purity. See protein gels in Figure 5 – figure supplement 1 showing purity of each MEG-3 variant.

MEG-3 protein as well as its fragments tend to aggregate and do not seem spherical as published previously (compare Fig. 2A from this manuscript with Fig. 4 and Fig 4S from Lee et al., 2020).

In the absence of RNA, MEG-3 form amorphous aggregates as reported in Lee et al., 2020. In the revised in vitro analyses, we avoid conditions that induce aggregates (e.g. high MEG-3 concentration/no RNA which is non physiological). We show that all the variants are soluble under high RNA/low salt conditions. High salt conditions in the presence of RNA lead to condensation and no aggregation. The new data are shown in Figures 5,6,7 and replace Figure 2 in the original version.

Protein quality control as well as performing gel shift assays, as in previous report, would indicate the quality of the protein and be important to resolve these inconsistencies. Partially aggregated MEG-3 might be present in the filter binding assays and eventually lead to much lower apparent RNA binding constants for the MEG-3 full length protein.

The conditions used in the filter binding assay (150mM NaCl and variable amounts of RNA) will prevent formation of MEG-3 aggregates, which only form in the absence of RNA (Lee et al., 2020).

Also, we now include new experiments that show that the apparent lower RNA affinity of full-length MEG-3 is only seen with poly-U30. Binding assays using *nos-2* RNA show no differences between full length MEG-3 and MEG-3 IDR (Figure 5).

In further in vitro assays it is tested whether MEG-3 interacts with PGL-3 condensates. Indeed, all MEG-3 fragments interact with the PGL condensates and form domains on the PGL condensates in the presence of RNA (see Fig. 2C). In the absence of RNA the MEG-3 IDR domain fully mixes with the PGL domain indicating strong interaction with PGL-3. However, this domain was not tested for interaction with PGL-3 by any other assays, and in vivo does not interact with PGLs.

Mixing into a condensate is NOT necessarily an indication of a strong or specific binding interaction. Phase separation is known to be driven by weak interactions and IDRs often show non-specific mixing. We have tested directly the IDR for binding to PGL-3 in a new bead assay (Figure 7) and found no evidence for specific binding to PGL-3.

Co-assembly of MEG/PGL condensates is driven by the MEG-3 C-terminus.The MEG-3 fragments tested in vitro are now generated in vivo by replacing the endogenous meg-3 locus with the gene variants by CRISPR/Cas9. The authors do a series of immunofluorescence experiments in the meg-3 variant lines and stain for MEG-3 and PGL-3. The authors state that “in embryos expressing MEG-3 Cterm, PGL-3 condensates localized properly in P1…”. However, as their images (see Fig4A) as well as their quantification indicates, this is not the case. Only full length MEG-3 shows proper localization of PGL-3, and neither the C-terminal, nor the IDR or the HMGL variant show correct PGL-3 localisation. Although there seems to be an enrichment of PGL-3 in P1 in the Cterm mutant, clearly other parts of the protein contribute to PGL-3 localization as seen in MEG-3 full length (see Fig. 4C)

We agree and have revised our discussion of what drives MEG and PGL asymmetry in the Discussion.

Efficient recruitment of mRNA to P granules requires all parts of the MEG-3 protein. Previously, the Seydoux lab discovered that hundreds of mRNA crosslink to MEG-3 in vivo and several of these mRNA are indeed enriched in the P lineage (Lee et al., 2020). Now, they test which part of MEG-3 is required for enrichment of Y51F10.2. Intriguingly, Y51F10.2 becomes only considerably enriched in P4 cells if all domains of the MEG-3 protein get expressed, indicating that enrichment of PGL’s by the MEG-3Cterm construct is not sufficient for enrichment of Y51F10.2 mRNA. However, to address if mRNA recruitment requires indeed all MEG-3 domains it would be much better to quantify the level of polyA enrichment in the different MEG-3 mutants (see Fig 5 supp1).

We have included an additional specific RNA probe (nos-2) in Figure 4 – Figure supplement 2, as well as quantifying the polyA data now in Figure 4 Figure supplement-3 to further support our conclusion that enrichment of mRNA in the P cell requires all the MEG-3 domains.

Previously, Lee et al have shown that nearly all polyA RNA which is accessible to FISH in early embryos is located in the P lineage in the P2 cell and this depends on the presence of wildtype copies of meg-3 and meg-4 (see Fig 1F, Lee et al., 2020). And if indeed the IDR of MEG-3 recruits mRNA than the MEG-3Cterm mutant should not enrich any mRNA in P1 to P4. Importantly, very little polyA RNA is enriched in pgl-1 pgl-3 mutants indicating that MEG-3 enrichment is not sufficient for P lineage enrichment of mRNA.

We agree and have clarified this in the text (see in particular last section of Discussion). All domains of MEG-3 are required for RNA enrichment.